# ELMUR: External Layer Memory with Update/Rewrite for Long-Horizon RL Problems

**Egor Cherepanov**[1,2]**, Alexey K. Kovalev**[1,2]**, Aleksandr I. Panov**[1,2]
[1]AXXX, [2]MIRIAI
`cherepanov@axxx.tech`

## Abstract

Real-world robotic agents must act under partial observability and long horizons, where key cues may appear long before they affect decision making. However, most modern approaches rely solely on instantaneous information, without incorporating insights from the past. Standard recurrent or transformer models struggle with retaining and leveraging long-term dependencies: context windows truncate history, while naive memory extensions fail under scale and sparsity. We propose **ELMUR** (**E**xternal **L**ayer **M**emory with **U**pdate/**R**ewrite), a transformer architecture with structured external memory. Each layer maintains memory embeddings, interacts with them via bidirectional cross-attention, and updates them through an **L**east **R**ecently **U**sed (**LRU**) memory module using replacement or convex blending. ELMUR extends effective horizons up to 100,000 times beyond the attention window and achieves a 100% success rate on a synthetic T-Maze task with corridors up to one million steps. In POPGym, it outperforms baselines on more than half of the tasks. On MIKASA-Robo sparse-reward manipulation tasks with visual observations, it nearly doubles the performance of strong baselines, achieving the best success rate on 21 out of 23 tasks and improving the aggregate success rate across all tasks by about 70% over the previous best baseline. These results demonstrate that structured, layer-local external memory offers a simple and scalable approach to decision making under partial observability. Code and project page: **https://elmur-paper.github.io/**.

## 1 Introduction

Imagine a robot cooking pasta: it stirs once, adds salt, and later adds salt again, repeating until the dish is inedible. The issue is simple: the robot cannot remember if salt was already added, since it dissolves invisibly, nor how much is still in the container. This is a case of partial observability — the world rarely reveals all necessary information. Humans recall past actions effortlessly, but robots lack this ability. Though effective in controlled settings (Kim et al., 2024; Black et al., 2024), robots often fail under partial observability (Fang et al., 2025; Cherepanov et al., 2026a). Standard recurrent (Ouyang et al., 2025) and transformer (Gao et al., 2025) models rely heavily on short observation windows, making them brittle under long-horizon dependencies and sparse signals. This motivates hybrid memory-augmented transformers that explicitly store and retrieve past information (Fang et al., 2025; Shi et al., 2025).

Within the Reinforcement Learning (RL) paradigm (Sutton et al., 1998), long-horizon challenges are compounded by sample inefficiency and sparse rewards: real-world exploration is costly and unsafe, while simulation suffers from a sim-to-real gap (Zhang et al., 2025a). Offline RL mitigates this with pre-collected datasets (Levine et al., 2020), but usually assumes dense feedback; reshaping sparse rewards demands domain knowledge and risks bias (Wu et al., 2021; Mu et al., 2024; Wang et al., 2025a). In robotics, delayed feedback makes long-term memory indispensable. A complementary paradigm is Imitation Learning (IL) (Zare et al., 2024), whose simplest form, Behavior Cloning (BC), reduces control to supervised learning on demonstration pairs. Building on this idea, recent Vision-Language-Action (VLA) models (Brohan et al., 2022; Team et al., 2024; Kim et al., 2024) scale with large datasets, yet their fixed transformer windows (Fang et al., 2025; Shi et al., 2025) leave three challenges: (i) extending context without quadratic cost, (ii) mitigating truncation-induced forgetting, and (iii) retaining task-relevant information across long horizons. This motivates our central question: **how can we equip IL policies with efficient long-term memory to solve long-horizon, partially observable tasks?**

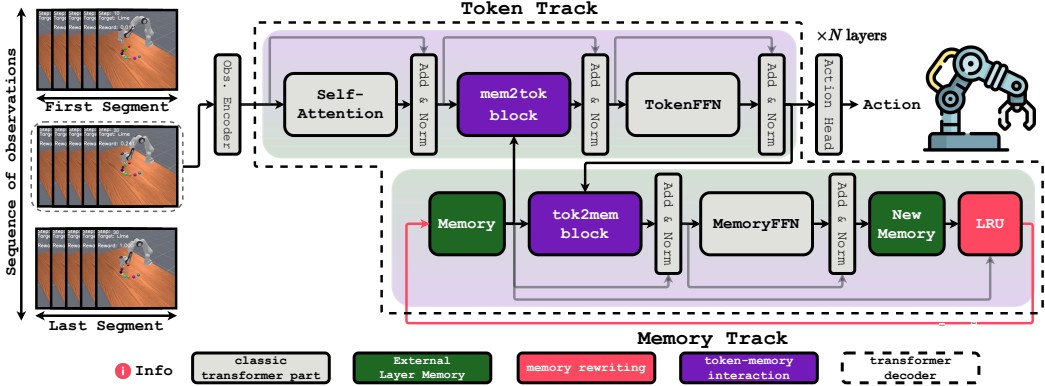

Figure 1: **ELMUR overview.** Each transformer layer is augmented with an external memory track that runs in parallel with the token track. Tokens attend to memory through a `mem2tok` block, while memory embeddings are updated from tokens through a `tok2mem` block. LRU block selectively rewrites memory via replacement or convex blending, ensuring bounded yet persistent storage. This design enables token-memory interaction and long-horizon recall beyond the attention window.

To address these challenges, we introduce **ELMUR** (**E**xternal **L**ayer **M**emory with **U**pdate/**R**ewrite), a transformer architecture in which every layer is augmented with a structured external layer memory (Figure 1). ELMUR combines three ingredients: (i) layer-local memory embeddings that persist across segments, (ii) bidirectional token–memory read/write interaction via cross-attention (`mem2tok`, `tok2mem`), and (iii) a Least Recently Used (LRU) update block that refreshes memory through replacement or convex blending, balancing stability and adaptability. This design enables efficient segment-level recurrence and extends retention of task-relevant information up to $100{,}000\times$ beyond the native attention window, making long-horizon decision making feasible in robotics.

We evaluate ELMUR on the synthetic T-Maze (Ni et al., 2023), the robotic MIKASA-Robo (Cherepanov et al., 2026a) suite of sparse-reward manipulation tasks with visual observations, and the diverse POPGym benchmark (Morad et al., 2023a), all designed to test memory under partial observability. ELMUR achieves a 100% success rate on T-Maze corridors up to one million steps, nearly doubles baseline performance on MIKASA-Robo, ranking first on 21 of 23 tasks and increasing the overall success rate across the suite by roughly 70% relative to the strongest prior method, and obtains the top score on 24 of 48 POPGym tasks. These results demonstrate that EL-MUR enables stable retention of task-relevant information, efficient long-term storage, and robust generalization under partial observability.

Our contributions are twofold:

- **We propose ELMUR**, a transformer with layer-local external memory, bidirectional token-memory cross-attention, and an LRU-based update rule rewriting memory via replacement or convex blending (Section 3). This design extends memory horizons far beyond the attention window.

- **We empirically demonstrate** that ELMUR achieves robust generalization under partial observability across synthetic, robotic, and puzzle/control tasks (Section 5).

- **We provide a theoretical analysis** of LRU-based memory dynamics, establishing formal bounds on forgetting, retention horizons, and stability of memory embeddings (Section 4).

## 2 BACKGROUND

Many real-world robotic and control tasks involve partial observability, where the agent cannot directly access the true system state (Lauri et al., 2022). This setting is modeled as a partially observable Markov decision process (POMDP), defined as the tuple $(\mathcal{S}, \mathcal{A}, \mathcal{O}, T, Z, R, \rho_0, \gamma)$, with latent state space $\mathcal{S}$, action space $\mathcal{A}$, and observation space $\mathcal{O}$. The transition dynamics are $T : \mathcal{S} \times \mathcal{A} \to \Delta(\mathcal{S})$, where $T(s' \mid s, a)$ is the probability of reaching $s'$ after taking $a$ in $s$. The observation function $Z : \mathcal{S} \times \mathcal{A} \to \Delta(\mathcal{O})$ specifies $Z(o \mid s', a)$, the probability of observing $o$ after

---

**Algorithm 1** ELMUR layer update for segment $i$ at layer $\ell$. Inputs are token hidden states $h \in \mathbb{R}^{B \times L \times d}$, memory $(m, p)$ with $m \in \mathbb{R}^{B \times M \times d}$, anchors $p \in \mathbb{Z}^{B \times M}$, and absolute times $t$. Outputs are updated hidden states $h'$ and memory $(m', p')$.

---

    *// Input embedding (before first layer) – to encode observation*
1: $h \leftarrow \text{ObsEncoder}(o)$
    *// Token track – sequence processing and enrichment with information from memory*
2: $h \leftarrow \text{AddNorm}(h + \text{SelfAttention}(h; \text{causal mask}))$
3: $B_{\text{rel}} \leftarrow \text{RelativeBias}(t, p)$   *// bias for adding temporal dependence*
4: $h \leftarrow \text{AddNorm}(h + \text{CrossAttention}(Q{=}h, K{=}m, V{=}m; \text{noncausal mask}, B_{\text{read}}))$
5: $h \leftarrow \text{AddNorm}(h + \text{TokenFFN}(h))$
6: $h' \leftarrow h$
    *// Output decoding (after final layer)*
7: $a \leftarrow \text{ActionHead}(h')$   *// map to action distribution and compute loss*
    *// Memory track*
8: $B_{\text{rel}} \leftarrow \text{RelativeBias}(p, t)$   *// reversed bias for write*
9: $u \leftarrow \text{AddNorm}(m + \text{CrossAttention}(Q{=}m, K{=}h', V{=}h'; \text{noncausal mask}, B_{\text{write}}))$
10: $\tilde{u} \leftarrow \text{AddNorm}(u + \text{MemoryFFN}(u))$
11: $(m', p') \leftarrow \text{LRU}(m, p, \tilde{u}, t)$
12: **return** $h', (m', p')$

---

reaching $s'$ under action $a$. The reward function is $R : \mathcal{S} \times \mathcal{A} \to \mathbb{R}$, the initial state distribution is $\rho_0 \in \Delta(\mathcal{S})$, and $\gamma \in (0, 1)$ is the discount factor.

In the special case of full observability, the observation equals the state ($o_t \equiv s_t$), reducing the POMDP to a Markov decision process (MDP). The optimal policy then depends only on the current state, $\pi^*(a_t \mid s_t)$. In the general POMDP case, however, the agent cannot access $s_t$ directly and must rely on the full history $h_t = (o_0, a_0, o_1, a_1, \ldots, o_t)$, yielding $\pi^*(a_t \mid h_t)$. A practical alternative is to approximate history with a learned memory state $m_t = f_\phi(m_{t-1}, o_t, a_{t-1})$,   $\pi_\theta : \mathcal{M} \to \Delta(\mathcal{A})$,   $\pi_\theta(a_t \mid m_t)$, where $f_\phi$ is a, for instance, recurrent (Hausknecht & Stone, 2015) or memory-augmented (Parisotto et al., 2020) update rule.

## 3   METHOD

Many real-world decision-making tasks involve long horizons and partial observability, where key information may appear thousands of steps before it is needed. Standard transformers are limited by a fixed attention window: naive extensions of context length increase cost quadratically, while truncation causes forgetting. Efficient long-term reasoning thus requires a mechanism to store and retrieve task-relevant information across long trajectories. To this end, we propose **ELMUR** (External Layer Memory with Update/Rewrite), a GPT-style (Radford et al., 2019) transformer decoder augmented with structured external memory. Unlike architectures that simply cache hidden states (Dai et al., 2019), ELMUR equips each layer with its own memory track and explicit read–write operations, enabling persistent storage and selective updating via the LRU memory management.

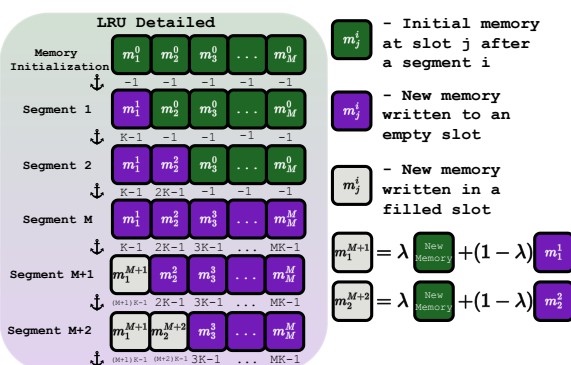

Figure 2: **LRU-based memory management in ELMUR.** Each layer maintains $M$ memory slots, initialized with random vectors (green). As new segments arrive, tokens write updates into empty slots (purple) by full replacement. Once all slots are filled, the least recently used slot is refreshed via a convex update with parameter $\lambda$ that blends new content with the previous memory (grey). Anchors below each row indicate the timestep of the most recent update. This scheme ensures bounded capacity while preserving long-horizon information.

**ELMUR Overview.** As shown in Figure 1, each ELMUR layer has two coupled tracks. The **token track** processes observations into actions, while the **memory track** persists across segments. Both interact through cross-attention: memory shapes token representations, and tokens update memory. Interaction occurs via `mem2tok` (read) and `tok2mem` (write) blocks, modulated by relative biases from token timesteps and memory anchors. Trajectories are split into segments, processed sequentially for efficiency and recurrent memory updates. At each segment's end, hidden states update memory, carried forward. LRU memory management fills empty slots first, then refreshes the least used slot by convexly blending old and new information. This bidirectional design provides temporally grounded memory for long-horizon decisions. Algorithm 1 summarizes the method.

**Segment-Level Recurrence.** Feeding infinitely long sequences into a transformer is infeasible, since self-attention scales quadratically. Splitting into shorter segments reduces cost but complicates information flow. Segment-level recurrence addresses this by treating the transformer as an RNN over segments, passing memory from one segment to the next (Dai et al., 2019; Bulatov et al., 2022). In ELMUR, this memory is realized as layer-local external memory instead of cached activations. Each layer maintains memory that is read within the current segment and updated before moving to the next. Formally, with context length $L$, a trajectory of length $T$ is partitioned into $S = \lceil T/L \rceil$ segments $\mathcal{S}_i$: $\mathbf{h}^{(i)} = \text{TokenTrack}(\mathcal{S}_i, \text{sg}(\mathbf{m}^{i-1}))$, where $\mathbf{h}^{(i)} \in \mathbb{R}^{B \times L \times d}$ denotes the hidden states of tokens in segment $i$, computed from the segment $\mathcal{S}_i$ and the detached memory $\text{sg}(\mathbf{m}^{i-1})$ carried from the previous segment.

**Token Track.** Within each segment $\mathcal{S}_i$, observations are encoded into token embeddings $\mathbf{x} \in \mathbb{R}^{L \times d}$, where $d$ is the model dimension. The token track models local dependencies and augments them with information from memory $\mathbf{m} \in \mathbb{R}^{M \times d}$. Standard transformers rely on fixed-window self-attention, whereas ELMUR also retrieves information from its external memory via cross-attention, allowing predictions to depend not only on recent tokens but on distant past events stored in memory. Self-attention, equipped with relative positional encodings (Dai et al., 2019) and a causal mask, models local dependencies within the segment:

$$\mathbf{h}_{\text{sa}} = \text{AddNorm}(\mathbf{x} + \text{SelfAttention}(\mathbf{x})), \tag{1}$$

where $\text{AddNorm}(\cdot)$ denotes a residual connection followed by normalization. Long-term context is handled by external memory. Tokens' hidden states $\mathbf{h}_{\text{sa}}$ then query memory via the `mem2tok`:

$$\mathbf{h}_{\text{mem2tok}} = \text{AddNorm}(\mathbf{h}_{\text{sa}} + \text{CrossAttention}(Q = \mathbf{h}_{\text{sa}}, K, V = \mathbf{m})). \tag{2}$$

Here memory embeddings act as keys and values, with a non-causal mask and a relative bias reflecting token-memory temporal distance. Finally, representations are refined with a feed-forward network (FFN). In contrast to popular Decision Transformer (DT) Chen et al. (2021) that employ a standard MLP-based FFN, we adopt a DeepSeek-MoE FFN (Dai et al., 2024), following the design of DeepSeek-V3 (Liu et al., 2024a). Mixture-of-Experts (MoE) improve parameter efficiency and specialization by routing tokens to a sparse set of experts, scaling capacity without proportional compute. This design enables expressive updates while keeping inference efficient:

$$\mathbf{h} = \text{AddNorm}(\mathbf{h}_{\text{mem2tok}} + \text{FFN}(\mathbf{h}_{\text{mem2tok}})). \tag{3}$$

The resulting hidden states are then passed to the action head, applied only after the final layer. Training is supervised, minimizing the error between predicted and demonstrated actions, using mean squared loss for continuous spaces and cross-entropy for discrete ones. The loss backpropagates through the entire network to update model parameters.

**Memory Track.** Reading from memory is not enough for long-horizon reasoning; the model must also write new information. Without an explicit write path, past events would be forgotten or cached inefficiently. The memory track addresses this by allowing tokens to update persistent memory, retaining salient information while overwriting less useful content.

Each layer maintains its own memory embeddings $\mathbf{m} \in \mathbb{R}^{M \times d}$. After processing a segment, token states update memory through the `tok2mem` block:

$$\mathbf{m}_{\text{tok2mem}} = \text{AddNorm}(\mathbf{m} + \text{CrossAttention}(Q = \mathbf{m}, K, V = \mathbf{h})). \tag{4}$$

As in `mem2tok`, a non-causal mask is applied, but the relative bias is reversed to favor temporally aligned memory embeddings. Updates are then refined by a FFN with residual connection:

$$\mathbf{m}_{\text{new}} = \text{AddNorm}(\mathbf{m}_{\text{tok2mem}} + \text{FFN}(\mathbf{m}_{\text{tok2mem}})), \tag{5}$$

analogous to the token track, the FFN uses a DeepSeek-MoE block instead of a standard MLP.

Finally, $\mathbf{m}_{\text{new}}$ is merged with existing slots via the LRU rule (Figure 2, Figure 2), filling empty slots first and otherwise refreshing the least recently used by convex blending. This keeps memory bounded yet consistently updated with relevant information.

**Relative Bias.** When memory extends across multiple segments, absolute indices become ambiguous: the same token position may correspond to different points in the trajectory. To resolve this, the model requires a signal that encodes relative distances between tokens and memory entries. ELMUR provides this signal through a learned relative bias added to cross-attention logits:

$$\text{Attn}(\mathbf{Q}, \mathbf{K}) = \frac{\mathbf{Q}\mathbf{K}^{\top}}{\sqrt{d_h}} + \mathbf{B}_{\text{rel}}. \tag{6}$$

The bias $\mathbf{B}_{\text{rel}}$ is derived from pairwise offsets $\Delta = \pm(t-p)$ between a token position $t$ and a memory anchor $p$ (the last update time of a slot). Offsets are clamped to $[-D_{\max}+1, D_{\max}-1]$, where $D_{\max}$ is the maximum relative distance supported by the bias table. These clamped values are shifted into $[0, 2D_{\max}-2]$ and used to index a learnable embedding table $\mathbf{E} \in \mathbb{R}^{(2D_{\max}-1)\times H}$, where $H$ is the number of attention heads. Each offset corresponds to a per-head embedding $\mathbf{E}[\Delta] \in \mathbb{R}^H$, and stacking these indices produces

$$\mathbf{B}_{\text{rel}} = \begin{cases} \mathbf{E}[t - p] \in \mathbb{R}^{B\times H\times L\times M}, \texttt{mem2tok (read)} \\ \mathbf{E}[p - t] \in \mathbb{R}^{B\times H\times M\times L}, \texttt{tok2mem (write)}. \end{cases} \tag{7}$$

In the read path (`mem2tok`), the bias prioritizes retrieval from temporally close memory embeddings while keeping distant ones accessible. In the write path (`tok2mem`), offsets are reversed, guiding updates toward memory embeddings aligned with the writing tokens. Both directions draw from the same embedding table $\mathbf{E}$ but can learn distinct patterns. By relying on relative rather than absolute timestep, ELMUR ensures consistent and coherent memory interactions across long horizons.

**Memory Management with LRU.** External memory must remain bounded: storing every token is infeasible, while naive truncation risks catastrophic forgetting. A principled policy is needed to decide which slots to refresh or preserve as new content arrives. ELMUR employs a Least Recently Used (LRU) block (Figure 2, Figure 2) that manages $M$ slots per layer, each holding a vector and an anchor (its last update time). By always updating the least recently used slot, the block ensures bounded capacity while retaining context.

---

**Algorithm 2** LRU update for layer memory. Inputs: current memory $(m, p)$ (may be uninitialized), candidate updates $\tilde{u}$, newest segment time $t$, blend $\lambda \in [0, 1]$, init scale $\sigma$. Output: updated memory $(m', p')$.

---

*// Initialization (cold start)*
1: **if** $m, p$ uninitialized **then**
2:    $m \leftarrow \mathcal{N}(0, \sigma^2 I)$   *// initial slots*
3:    $p \leftarrow -\mathbf{1}$   *// sentinel anchors*
4: **end if**
   *// Choose write index*
5: empty $\leftarrow (p < 0)$
6: **if** any empty **then**
7:    $j^\star \leftarrow \text{first}(\text{empty})$   *// use first empty slot*
8:    $\alpha \leftarrow 1$   *// full replacement*
9: **else**
10:    $j^\star \leftarrow \arg\min_j p_j$   *// least recently used*
11:    $\alpha \leftarrow \lambda$   *// convex blend*
12: **end if**
   *// Integrate*
13: blend $\leftarrow \alpha \tilde{u}_{j^\star} + (1 - \alpha) m_{j^\star}$
14: $m' \leftarrow m$;  $m'_{j^\star} \leftarrow$ blend
15: $p' \leftarrow p$;  $p'_{j^\star} \leftarrow t$
16: **return** $(m', p')$

---

At training start, **initialization** samples embeddings from $\mathcal{N}(0, \sigma^2 I)$ and marks them empty. While empty slots remain, **full replacement** inserts new vectors directly. Once all slots are filled, the block switches to convex update, blending the oldest slot with new content:

$$\mathbf{m}_j^{i+1} = \lambda \mathbf{m}_{\text{new}}^{i+1} + (1 - \lambda) \mathbf{m}_j^i, \tag{8}$$

where $\lambda \in [0, 1]$ is a tunable hyperparameter that controls the balance between overwriting and retention. By adjusting $\lambda$, one can choose whether memory favors fast plasticity (larger $\lambda$) or long-term stability (smaller $\lambda$). This policy uses memory capacity fully before overwriting and applies gradual blending thereafter, enabling bounded yet persistent long-horizon memory.

By combining token-level processing with an explicit memory system, ELMUR offers three core advantages: (i) relative-bias cross-attention provides temporally grounded read-write access, (ii) the LRU-based manager ensures bounded capacity while remaining adaptive, and (iii) segment-level recurrence enables scalable learning over long horizons.

## 4 THEORETICAL ANALYSIS

Understanding the retention properties of ELMUR's memory is crucial for characterizing its ability to handle long-horizon dependencies. In this section, we analyze how information is preserved or forgotten under the LRU update mechanism. We derive bounds on memory retention and effective horizons, and connect these results to the empirical behaviors observed in long-horizon tasks.

At the core of ELMUR's memory module is the convex update rule with blending factor $\lambda \in [0, 1]$. Let fix a memory embedding $j$ at segment index $i$. If this memory embedding is selected for update with new content $\mathbf{m}_{\text{new}}^{i+1}$, the rule (Figure 2) is $\mathbf{m}_j^{i+1} = \lambda \mathbf{m}_{\text{new}}^{i+1} + (1 - \lambda) \mathbf{m}_j^i$, while all other memory embeddings $n \neq j$ remain unchanged: $\mathbf{m}_n^{i+1} = \mathbf{m}_n^i$. If the memory embedding was empty, the update reduces to full replacement $\mathbf{m}_j^{i+1} = \mathbf{m}_{\text{new}}^{i+1}$.

**Proposition 1 (Exponential Forgetting).** After $k$ overwrites of memory embedding $j$, the content evolves as

$$\mathbf{m}_j^{i+k} = (1 - \lambda)^k \mathbf{m}_j^i + \sum_{u=1}^{k} \lambda(1 - \lambda)^{k-u} \mathbf{m}_{\text{new}}^{i+u}, \tag{9}$$

where $\mathbf{m}_{\text{new}}^{i+u}$ denotes the write at update $i+u$ (see Appendix A.1 for a full derivation). Consequently, the coefficient of the initial content $\mathbf{m}_j^i$ after $k$ overwrites is $(1 - \lambda)^k$, and the contribution of the write performed $\tau$ updates earlier is $\lambda(1 - \lambda)^{\tau-1}$.

**Corollary (Half-life).** The number of overwrites $k_{0.5}$ after which the contribution of $\mathbf{m}_j^i$ halves is $k_{0.5} = \frac{\ln(1/2)}{\ln(1-\lambda)} = \frac{\ln 2}{-\ln(1-\lambda)} \sim \frac{\ln 2}{\lambda}$, as $\lambda \to 0$. Thus, smaller $\lambda$ extends retention, while larger $\lambda$ accelerates overwriting.

**Effective horizon in environment steps.** Since only one memory embedding is updated per segment of length $L$, a memory is overwritten once every $M$ segments in expectation. The *effective retention horizon* $H(\epsilon)$ thus quantifies how many environment steps a stored contribution remains influential before its weight decays below a negligible threshold $\epsilon$, i.e., $H(\epsilon) = M \cdot L \cdot \frac{\ln(\epsilon)}{\ln(1-\lambda)}$. In particular, the half-life in environment steps is $H_{0.5} = M \cdot L \cdot \frac{\ln 2}{-\ln(1-\lambda)} \sim M \cdot L \cdot \frac{\ln 2}{\lambda}$, as $\lambda \to 0$.

Unlike models where all memory is updated at every step (like RNNs), ELMUR's LRU policy ensures (i) memory embeddings not selected for overwrite retain their content exactly until replacement, and (ii) once selected, their contributions decay exponentially with rate $\lambda$. This produces a retention horizon that scales linearly with both the number of memory embeddings $M$ and the segment length $L$, providing a conservative lower bound. In practice, effective horizons are often much longer (Figure 3).

**Proposition 2 (Memory Boundedness).** A natural question is whether repeated convex updates could cause memory values to grow without limit. We show that, under standard bounded-input assumptions, the norm of every memory embedding remains uniformly bounded throughout training and inference. Suppose that every new write is norm-bounded, $\|\mathbf{m}_{\text{new}}^t\| \leq C$ for some constant $C > 0$, and the initial memory satisfies $\|\mathbf{m}_j^0\| \leq C$. Then for all segments $i$ and slots $j$, it holds that $\|\mathbf{m}_j^i\| \leq C$. Since each update is a convex combination of the previous and a bounded new values, the memory embedding always remains inside the closed ball of radius $C$. This guarantees stability of activations even across arbitrarily long trajectories. See Appendix A.2 for the detailed proof.

## 5 EXPERIMENTS

We evaluate ELMUR on synthetic (Ni et al., 2023) tasks, 48 POPGym puzzle/control tasks (Morad et al., 2023a), and robotic manipulation (Cherepanov et al., 2026a), all designed to test memory under partial observability. Our study is guided by the following research questions (RQs):

1. **RQ1**: Does ELMUR retain information across horizons far beyond its attention window?
2. **RQ2**: How well does ELMUR generalize to shorter and longer sequences?
3. **RQ3**: Is ELMUR effective on manipulation tasks with visual observations?
4. **RQ4**: How consistent is ELMUR across puzzles, control, and robotics tasks?
5. **RQ5**: What is the impact of components of ELMUR on its memorization?

### 5.1 BENCHMARKS AND BASELINES

We evaluate ELMUR on three benchmarks designed to isolate memory (Appendix, Figure 7). The **T-Maze** requires recalling an early cue after traversing a long corridor with sparse rewards. The **MIKASA-Robo** suite provides robotic tabletop tasks with RGB observations and continuous actions, including color-recall (`RememberColor`) and delayed reversal (`TakeItBack`). Finally, **POPGym** offers a diverse collection of partially observable puzzles and control environments for evaluating general memory use. Detailed descriptions can be found in Appendix A.4.

We compare against baselines spanning sequence models and offline RL for long-horizon tasks. We include transformers — Decision Transformer (DT) (Chen et al., 2021) and Recurrent Action Transformer with Memory (RATE) (Cherepanov et al., 2026c) — as representative architectures for memory-augmented policy learning. We also evaluate DMamba (Ota, 2024), a state-space model with efficient recurrence, as a recent alternative to attention. For IL/offline RL, we use Behavior Cloning (BC) via MLP as the simplest supervised baseline, Conservative Q-Learning (CQL) (Kumar et al., 2020) as a strong offline RL method, and Diffusion Policy (DP) (Chi et al., 2023) as a state-of-the-art generative policy. Together, these span transformer, state-space, and offline/generative approaches, providing a competitive reference set for evaluation. We do not compare with online RL baselines, since they assume interactive data collection with exploration, yielding incomparable training budgets. Likewise, we omit real-robot experiments to avoid confounds such as latency, resets, and safety constraints, focusing instead on controlled, reproducible studies.

**Experimental Setup.** For **RQ1**, we test T-Maze cue retention by training with short contexts ($L$=10, $S$=3) and evaluating on corridors up to $10^6$ steps. For **RQ2**, we train on 7 T-Maze lengths distributions (9–900) and validate on 11 shorter/longer ones (9–9600) to assess interpolation and extrapolation. For **RQ3**, we use MIKASA-Robo tasks, training by imitation from expert demonstrations and evaluating zero-shot. For **RQ4**, we compare T-Maze, POPGym-48, and MIKASA-Robo to test robustness across synthetic puzzles, control, and robotics. For **RQ5**, we ablate `RememberColor3-v0`, varying $M$, $\lambda$, $\sigma$, and $(L, S)$, and remove relative bias, LRU, and per-layer memory to measure component contributions.

**Evaluation Protocol.** Unless stated otherwise, each model is trained with three (four for T-Maze) independent runs (different initialization). For each run we evaluate on 100 episodes with distinct environment seeds and compute the run mean. We then report the grand mean $\pm$ standard error of the mean (SEM) across the three run means. For per-task leaderboards (e.g., POPGym-48) we apply this protocol per task and aggregate as specified in the benchmark.

**Training Details and Hardware.** All models are trained from scratch under the same data budgets and preprocessing. We use segment-level recurrence with detached memory between segments; losses are applied on each processed segment. Optimizers, schedulers, and hyperparameters follow the task-specific configuration table in Appendix, Table 7. All experiments were run on a single NVIDIA A100 (80 GB) per job. Training/evaluation code paths, seeds, and environment versions are fixed across methods for reproducibility.

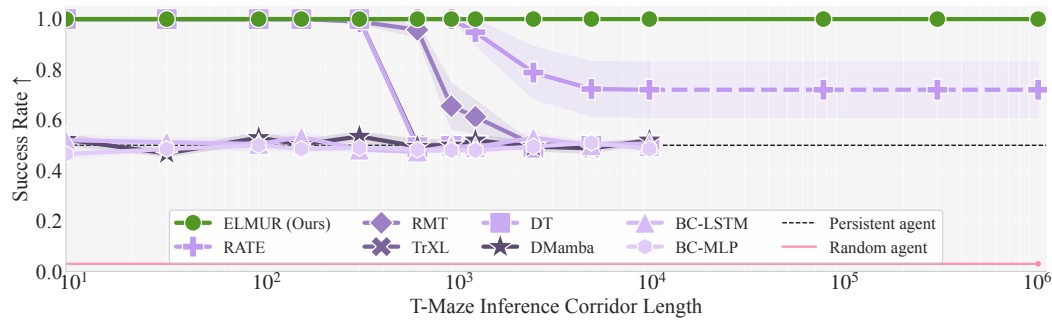

Figure 3: Success rate on the T-Maze task as a function of inference corridor length. ELMUR achieves a **100% success rate** up to corridor lengths of **one million steps**. In this figure, the context length is $L = 10$ with $S = 3$ segments; thus **ELMUR carries information across horizons 100,000 times longer than its context window**.

## 5.2 RESULTS

We evaluate ELMUR on T-Maze, MIKASA-Robo, and POPGym, addressing RQ1–RQ5 on retention, generalization, manipulation, cross-domain robustness, and ablations.

**RQ1: Retention beyond attention.** To test memory retention, we train on T-Maze corridors of length $T$ while restricting the context size to $L < T$, forcing the model to solve tasks where the cue must be preserved beyond the native attention span. At validation, we evaluate on much longer

corridors — up to one million steps — without increasing $L$, thereby probing memory retention far beyond the training horizon. ELMUR

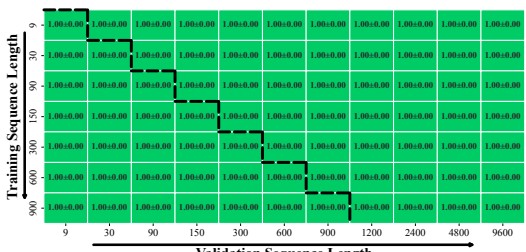

Figure 4: **Generalization of ELMUR across T-Maze lengths.** Each cell shows success rate (mean ± standard error) for training vs. validation lengths. ELMUR transfers perfectly: models trained on shorter sequences retain 100% success up to 9600 steps. Training lengths were split into three equal segments.

achieves 100% success even under this extreme extrapolation (Figure 3), implying retention horizons nearly 100,000× larger than the attention window ($L$=10 with only $S$=3 segments used during training).

**RQ2: Generalization across sequence lengths.** We train ELMUR on T-Maze with short contexts (3 to 300 steps) and then evaluate across 11 validation lengths ranging from 9 to 9600 steps. The model transfers seamlessly in both directions: it solves tasks shorter than those seen during training without overfitting to a fixed scale, and it also extrapolates to sequences orders of magnitude longer. As shown in Figure 4, ELMUR maintains 100% success across all train/test pairs, demonstrating robust generalization beyond the training horizon.

**RQ3: Manipulation with visual observations.** Results in Table 1 indicate that EL-MUR achieves higher success rates than other baselines on the MIKASA-Robo tasks. In

Table 1: Success rates (mean ± standard error) on MIKASA-Robo tasks, averaged over 3 runs with 100 evaluation seeds. ELMUR outperforms baselines, showing stronger memory in manipulation. See results for all 32 MIKASA-Robo tasks in Appendix, Table 8

.

| Task | RATE | DT | BC-MLP | CQL-MLP | DP | ELMUR (ours) |
|------|------|-----|--------|---------|-----|--------------|
| RememberColor3-v0 | 0.65±0.04 | 0.01±0.01 | 0.27±0.03 | 0.29±0.01 | 0.32±0.01 | **0.89±0.07** |
| RememberColor5-v0 | 0.13±0.03 | 0.07±0.05 | 0.12±0.01 | 0.15±0.02 | 0.10±0.02 | **0.19±0.03** |
| RememberColor9-v0 | 0.09±0.02 | 0.01±0.01 | 0.12±0.02 | 0.15±0.01 | 0.17±0.01 | **0.23±0.02** |
| TakeItBack-v0 | 0.42±0.24 | 0.08±0.04 | 0.33±0.10 | 0.04±0.01 | 0.05±0.02 | **0.78±0.03** |

`TakeItBack-v0`, it obtains $0.78\pm0.03$ compared to $0.42\pm0.24$ for the next-best model, and in `RememberColor[3,5,9]-v0` its performance remains stable as the number of distractors increases. Overall, ELMUR

Table 2: Aggregated returns on 48 POPGym tasks.

|  | RATE | DT | Rand. | BC-MLP | BC-LSTM | ELMUR |
|---|---|---|---|---|---|---|
| **All (48)** | 9.5 | 5.8 | -12.2 | -6.8 | 9.0 | **10.4** |
| **Puzzle (33)** | 0.45 | -3.5 | -14.6 | -11.9 | -0.2 | **1.2** |
| **Reactive (15)** | 9.1 | 9.3 | 2.3 | 5.1 | 9.1 | 9.2 |

shows more reliable performance under visual interference in manipulation tasks with pixel inputs.

**RQ4: Robustness across domains.** Across synthetic (T-Maze), control/puzzle (48 POPGym), and robotic (MIKASA-Robo) benchmarks, ELMUR consistently outperforms baselines, generalizing across diverse modalities, actions, and rewards. On POPGym, it achieves the best overall score (10.4), with the largest gains on memory puzzles (1.2 vs. 0.45 for RATE; DT and BC-LSTM score below zero), showing the importance of explicit memory for long-term dependencies. On reactive tasks, ELMUR stays competitive without sacrificing puzzle performance, ranking first on 24 of 48 tasks (full results in Table 5). Figure 5 shows consistent per-task gains over DT, especially on memory-intensive puzzles. Improved retention comes with little overhead: on T-Maze, ELMUR has 2.1M parameters (vs. 1.7M for RATE, 1.8M for DT) yet runs faster per step ($6.8\pm0.5$ ms) than RATE ($7.2\pm0.3$ ms) and DT ($10.7\pm0.1$ ms). Efficiency stems from (i) a short attention window with long-term context handled by bounded memory, so complexity depends on memory size not sequence length, and (ii) MoE feed-forward layers, which raise capacity without proportional compute. Thus, explicit memory is both effective and efficient for long-horizon RL.

**RQ5: Ablation Study.** We ablate ELMUR's memory design on `RememberColor3-v0` (Figure 6, Table 3). Unless noted, models use *per-layer* memory, relative-bias token–memory cross-attention, and LRU-based updates; *shared memory* denotes embeddings shared across layers. In Figure 6 (b–d) the LRU factor is fixed to $\lambda = 0$ to isolate other effects. Results average three runs of 20 episodes. Performance scales with memory size $M$: when $M \geq N$ (the number of segments needed), success is near-perfect; when $M < N$, accuracy drops sharply, especially near

Table 3: Ablation study results.

| Setting | Score |
|---|---|
| Baseline ELMUR | $1.00 \pm 0.00$ |
| Shared memory | $0.45 \pm 0.03$ |
| No rel. bias | $0.95 \pm 0.05$ |
| No LRU | $0.43 \pm 0.22$ |
| No rel. bias; No LRU | $0.22 \pm 0.11$ |
| MoE $\rightarrow$ MLP | $1.00 \pm 0.00$ |

$M \approx N$ (Figure 6, c–d). Intermediate blending ($\lambda \approx 0.4$–$0.6$) is unstable (Figure 6, a), while larger initialization $\sigma$ mitigates collapse (Figure 6, b). Finer recurrence (shorter segments, larger $N$) stresses capacity unless $M$ scales accordingly. Component ablations confirm that capacity and LRU dominate. Removing LRU leaves stale entries, and removing both LRU and relative bias prevents effective retrieval. Relative bias gives modest gains, while shared memory degrades performance, underscoring the value of layer-local design. Finally, replacing MoE-FFN with MLP-FFN preserves accuracy while improving computational efficiency.

To confirm that memory mechanisms do not harm performance on fully observable MDPs, we evaluated all models on the simple control task `CartPole-v1` (Towers et al., 2024). ELMUR, RATE, RMT, TrXL, BC-MLP, BC-LSTM, and CQL all achieved the maximum return of $500 \pm 0$, showing that adding memory does not break performance in standard MDP settings. See additional results on more competitive D4RL benchmark (Todorov et al., 2012) with MDP tasks in the Appendix, Section A.5, Table 4.

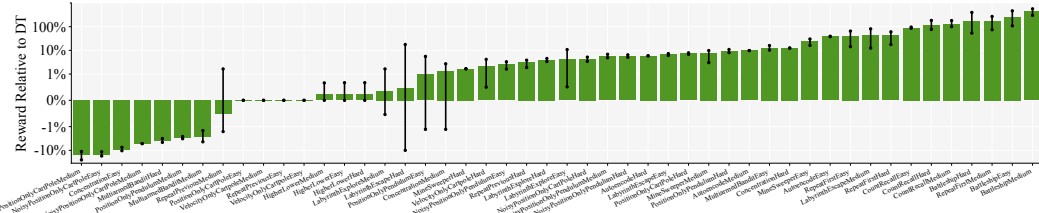

Figure 5: ELMUR compared to DT on all 48 POPGym tasks. Each model was trained with three independent runs, validated over 100 episodes each. Bars show the mean performance with 95% confidence intervals computed over these three means. ELMUR achieves consistent improvements over DT, with the largest gains on memory-intensive puzzles.

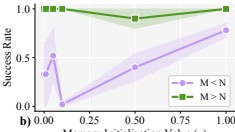 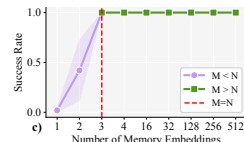 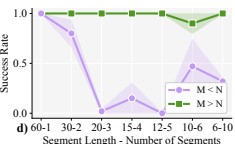

Figure 6: **Ablations of ELMUR's memory hyperparameters on `RememberColor3-v0`.** (a) LRU blending factor ($\lambda$), (b) memory embeddings initialization ($\sigma$), (c) number of memory embeddings ($M$), and (d) segment configuration ($L - S$). Curves compare settings where the number of memory embeddings is smaller than the required segments to solve the task ($M < N$) versus larger ($M > N$). Results show that sufficient memory capacity ($M \geq N$) yields stable success, while under-provisioned memory ($M < N$) is highly sensitive to $\lambda$, $\sigma$, and segmentation.

## 6 RELATED WORK

**Manipulation.** Transformer approaches to robotic manipulation can be broadly categorized by their underlying design principles. *Perception-centric visuomotor transformers* focus on multi-view or 3D perception to improve near fully observable control (Shridhar et al., 2023; Goyal et al., 2024). *Sequence/skill modeling* distills demonstrations into reusable action chunks but remains bottle-necked by limited context (Huang et al., 2023; Kobayashi et al., 2025). *Planning/value-augmented transformers* integrate transformers with planning or value learning for closed-loop control under finite context (Zhang et al., 2025b; Hu et al., 2025). *Alternative backbones* adopt state-space models or diffusion for efficiency, but without persistent memory (Liu et al., 2024b; Chi et al., 2023). *Scaling to VLA* broadens task coverage with language but still suffers from fixed horizons, with some remedies via summarization, feature banks, or hierarchy (Zitkovich et al., 2023; Team et al., 2024; Kim et al., 2024; Fang et al., 2025; Shi et al., 2025). ELMUR differs by training as a standard IL transformer while removing the context bottleneck through structured, layer-local external memory.

**Memory.** Efforts to extend sequence models to long horizons take several forms. *Implicit recurrence and state-space models* compress history in hidden dynamics, offering efficiency but little control over forgetting (Beck et al., 2024; Gu & Dao, 2023). *External memory with learned access* provides addressable storage but complicates optimization (Graves et al., 2016; Santoro et al., 2016). *Transformer context extension* retains history via caches or auxiliary slots but keeps memory peripheral (Dai et al., 2019). In RL, memory is often implemented through *episodic buffers* for salient events (Lampinen et al., 2021) or *sequence-model adaptations* that retrofit transformers for recurrence (Parisotto et al., 2020; Cherepanov et al., 2026b). Architectures vary in integration: RATE (Cherepanov et al., 2026c) concatenates memory with tokens, Memformer (Wu et al., 2020) uses global slots, and Block-Recurrent Transformers (Hutchins et al., 2022) recycle hidden states. ELMUR instead gives each layer an external memory with dedicated `mem2tok`/`tok2mem` cross-attention and LRU updates, yielding bounded memory for long-horizon tasks (Appendix A.6).

## 7 CONCLUSION

We introduced ELMUR, a transformer architecture with layer-local external memory, bidirectional token–memory cross-attention, and an LRU-based update rule. Unlike prior methods, ELMUR integrates explicit memory into every layer, achieving retention horizons up to $100,000\times$ beyond the native attention window. Our analysis establishes formal guarantees on half-life and boundedness under convex blending, and experiments on T-Maze, 48 POPGym tasks, and MIKASA-Robo demonstrate consistent improvements over strong baselines, underscoring reliable credit assignment under partial observability. We envision ELMUR as a simple and extensible framework for long-horizon decision-making with scalable memory in sequential control.

## REPRODUCIBILITY STATEMENT

We have taken several measures to ensure the reproducibility of our results. **Model details:** A complete description of the ELMUR architecture, including pseudocode for the layer update and the LRU-based memory module, is provided in Section 3, Algorithm 1, and Figure 2. **Theoretical re-**

**sults:** All assumptions and formal proofs of our propositions on exponential forgetting, half-life, and boundedness are presented in Section 4 and detailed in Appendix A.1–A.2. **Experimental setup:** Benchmarks, training procedures, and evaluation protocols are described in Section 5, with additional specifications (hyperparameters, dataset preprocessing, random seeds, and hardware setup) reported in Appendix 7 and A.4. **Baselines:** All baselines are implemented from open-source libraries or faithfully re-implemented with hyperparameters matched to their original publications, as described in Section 5 and Appendix A.6. **Code and data:** An anonymous repository with the implementation of ELMUR, training scripts, and configuration files is provided in the supplementary material. Together, these resources enable full replication of both our theoretical analysis and empirical findings.

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

## A APPENDIX

### A.1 PROOF OF EXPONENTIAL FORGETTING IN LRU UPDATES

**Proposition 1 (Exponential Forgetting).** Fix a slot $j$ and a segment index $i$. Suppose this slot is updated $k$ times at segments $i+1, \ldots, i+k$ according to

$$\mathbf{m}_j^t = \lambda \mathbf{m}_{\text{new}}^t + (1-\lambda) \mathbf{m}_j^{t-1}, \qquad t = i+1, \ldots, i+k, \tag{10}$$

with $0 \leq \lambda \leq 1$. Then

$$\mathbf{m}_j^{i+k} = (1-\lambda)^k \mathbf{m}_j^i + \sum_{u=1}^{k} \lambda(1-\lambda)^{k-u} \mathbf{m}_{\text{new}}^{i+u}. \tag{11}$$

Consequently, the coefficient of the initial content $\mathbf{m}_j^i$ after $k$ overwrites is $(1-\lambda)^k$, and the coefficient of the write performed $\tau$ updates ago (at segment $i+\tau$) is $\lambda(1-\lambda)^{\tau-1}$.

**Proof.** We prove Equation 11 by induction on $k$.

*Base case $k = 1$.* By the update rule,

$$\mathbf{m}_j^{i+1} = \lambda \mathbf{m}_{\text{new}}^{i+1} + (1-\lambda) \mathbf{m}_j^i, \tag{12}$$

which matches Equation 11 with $k = 1$.

*Induction step.* Assume Equation 11 holds for some $k \geq 1$. For $k+1$,

$$\mathbf{m}_j^{i+k+1} = \lambda \mathbf{m}_{\text{new}}^{i+k+1} + (1-\lambda) \mathbf{m}_j^{i+k}. \tag{13}$$

Insert the induction hypothesis for $\mathbf{m}_j^{i+k}$:

$$\mathbf{m}_j^{i+k+1} = \lambda \mathbf{m}_{\text{new}}^{i+k+1} + (1-\lambda) \left[ (1-\lambda)^k \mathbf{m}_j^i + \sum_{u=1}^{k} \lambda(1-\lambda)^{k-u} \mathbf{m}_{\text{new}}^{i+u} \right]. \tag{14}$$

Distribute $(1-\lambda)$ and regroup terms:

$$\mathbf{m}_j^{i+k+1} = (1-\lambda)^{k+1} \mathbf{m}_j^i + \sum_{u=1}^{k} \lambda(1-\lambda)^{(k+1)-u} \mathbf{m}_{\text{new}}^{i+u} + \lambda \mathbf{m}_{\text{new}}^{i+k+1}, \tag{15}$$

which is exactly Equation 11 with $k$ replaced by $k+1$. This completes the induction.

Finally, the coefficients in Equation 11 form a convex combination:

$$(1-\lambda)^k + \sum_{u=1}^{k} \lambda(1-\lambda)^{k-u} = (1-\lambda)^k + \lambda \sum_{r=0}^{k-1} (1-\lambda)^r = 1, \tag{16}$$

so it is meaningful to call them "fractions" of contribution. ∎

**Corollary (Half-life).** The number of overwrites $k_{0.5}$ after which the contribution of $\mathbf{m}_j^i$ halves satisfies

$$(1-\lambda)^{k_{0.5}} = \tfrac{1}{2} \implies k_{0.5} = \frac{\ln(1/2)}{\ln(1-\lambda)}. \tag{17}$$

Equivalently,

$$k_{0.5} = \frac{\ln 2}{-\ln(1-\lambda)}. \tag{18}$$

Using the Maclaurin series expansion $\ln(1-\lambda) \sim -\lambda$ as $\lambda \to 0$, we obtain

$$\lim_{\lambda \to 0} k_{0.5} \cdot \lambda = \ln 2. \tag{19}$$

Hence,

$$k_{0.5} \sim \frac{\ln 2}{\lambda} \quad \text{as } \lambda \to 0, \tag{20}$$

showing that smaller $\lambda$ yields longer retention horizons, while larger $\lambda$ overwrites past content more aggressively.

A.2 BOUNDEDNESS OF MEMORY EMBEDDINGS

Fix a slot $j$ and consider its update rule,

$$\mathbf{m}_j^{i+1} = \lambda \, \mathbf{m}_{\text{new}}^{i+1} + (1 - \lambda) \, \mathbf{m}_j^i, \qquad 0 \le \lambda \le 1. \tag{21}$$

We prove the claim by induction on $i$.

*Base case.* At $i = 0$, the assumption gives $\|\mathbf{m}_j^0\| \le C$.

*Induction step.* Assume $\|\mathbf{m}_j^i\| \le C$ for some $i \ge 0$. Using the update rule,

$$\|\mathbf{m}_j^{i+1}\| = \|\lambda \, \mathbf{m}_{\text{new}}^{i+1} + (1 - \lambda) \, \mathbf{m}_j^i\|. \tag{22}$$

By the triangle inequality and the inductive hypothesis,

$$\|\mathbf{m}_j^{i+1}\| \le \lambda \|\mathbf{m}_{\text{new}}^{i+1}\| + (1 - \lambda)\|\mathbf{m}_j^i\| \le \lambda C + (1 - \lambda)C = C. \tag{23}$$

Thus $\|\mathbf{m}_j^{i+1}\| \le C$, completing the induction.

Therefore, for all $i$, the memory embedding satisfies $\|\mathbf{m}_j^i\| \le C$. ∎

A.3 MEMORY-INTENSIVE ENVIRONMENTS

**Event–recall pairs and correlation horizon.** Following (Cherepanov et al., 2026b), let $\alpha_{t_e}^{\Delta t}$ denote an event of duration $\Delta t$ starting at time $t_e$, and let $\beta_{t_r}(\alpha_{t_e}^{\Delta t})$ be a later decision point that must recall information from that event. Define the **correlation horizon** as

$$\xi = t_r - t_e - \Delta t + 1. \tag{24}$$

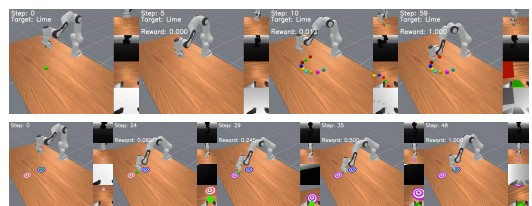

(a) MIKASA-Robo: `RememberColor9-v0` (top) and `TakeItBack-v0` (bottom).

For an environment, collect all event–recall horizons into the set $\Xi = \{\xi_n\}_n$.

Recent works have proposed alternative but complementary definitions of memory-intensive environments. For instance, Wang et al. (2025b) formalize **memory demand structures** that capture the minimal past sufficient to predict future transitions and rewards, while Yue et al. (2024) introduce **memory dependency pairs** to annotate which past observations must be recalled for correct decisions.

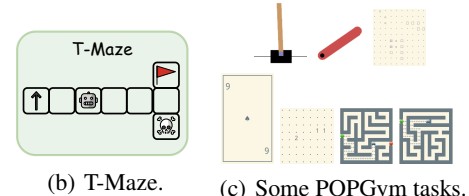

(b) T-Maze.  (c) Some POPGym tasks.

Figure 7: **Environments used in experiments.**

Both perspectives emphasize controllable ways to scale memory difficulty, either by increasing order/span or by manipulating dependency graphs. In this paper, we adopt the event–recall horizon framework for clarity and analytical tractability, but note that these alternative views are broadly consistent and provide useful tools for designing and benchmarking memory-intensive tasks.

**Definition (memory-intensive environment).** A POMDP $\tilde{M}_P$ is **memory-intensive** if $\min_n \Xi > 1$; i.e., every relevant decision depends on information separated by at least one intervening step, making reactive (myopic) policies insufficient. This definition cleanly separates POMDPs that genuinely require memory from those that are effectively MDP-like.

A.4 MEMORY-INTENSIVE ENVIRONMENTS

The memory-intensive environments used in this work are presented in Figure 7.

**T-Maze.** The T-Maze (Ni et al., 2023) features a corridor ending in a junction with two goals. At the start, one goal is randomly revealed, and the agent must recall this cue after traversing the corridor to choose the correct branch. Observations are vectors; actions are discrete. Rewards are sparse, provided only upon reaching the correct goal. The task tests whether the model can retain early cues across long delays.

Table 4: Normalized scores on MuJoCo tasks from the D4RL benchmark (Todorov et al., 2012). Baselines include Trajectory Transformer (TT) (Janner et al., 2021), Decision Transformer (DT) (Chen et al., 2021), Conservative Q-Learning (CQL) (Kumar et al., 2020), and Behavior Cloning (BC) from (Chen et al., 2021). **Top-1** and **Top-2** results are highlighted. $(R, o, a)$ and $(o)$ denote two conditioning modes of ELMUR: conditioning on returns-to-go, observations, and actions, or conditioning on observations alone, respectively.

| Dataset | Environment | CQL | DT | BC | TT | RATE | ELMUR $(R, o, a)$ | ELMUR $(o)$ |
|---------|-------------|-----|-----|-----|-----|------|-------------------|-------------|
| ME | HalfCheetah | 91.6 | 86.8±1.3 | 59.9 | 95.0±0.2 | 87.4±0.1 | 86.4±2.7 | 89.1±0.6 |
| ME | Hopper | 105.4 | 107.6±1.8 | 79.6 | 110.0±2.7 | 112.5±0.2 | 111.7±0.2 | 85.3±1.6 |
| ME | Walker2d | 108.8 | 108.1±0.2 | 36.6 | 101.9±6.8 | 108.7±0.5 | 108.9±0.2 | 108.4±0.1 |
| M | HalfCheetah | 44.4 | 42.6±0.1 | 43.1 | 46.9±0.4 | 43.5±0.3 | 43.2±0.1 | 43.1±0.1 |
| M | Hopper | 58.0 | 67.6±1.0 | 63.9 | 61.1±3.6 | 77.4±1.4 | 78.0±0.8 | 78.6±1.0 |
| M | Walker2d | 72.5 | 74.0±1.4 | 77.3 | 79.0±2.8 | 80.7±0.7 | 79.4±0.2 | 79.5±0.4 |
| MR | HalfCheetah | 45.5 | 36.6±0.8 | 4.3 | 41.9±2.5 | 39.0±0.6 | 40.4±0.6 | 39.8±1.0 |
| MR | Hopper | 95.0 | 82.7±7.0 | 27.6 | 91.5±3.6 | 83.7±8.2 | 64.8±1.6 | 87.6±1.7 |
| MR | Walker2d | 77.2 | 66.6±3.0 | 36.9 | 82.6±6.9 | 73.7±1.4 | 75.0±6.8 | 60.8±1.7 |
| | **Average** | 77.6 | 74.7 | 47.7 | 78.9 | 78.5 | 76.4 | 74.7 |

**MIKASA-Robo benchmark.** We further evaluate on the MIKASA-Robo benchmark (Cherepanov et al., 2026a), which provides robotic tabletop manipulation tasks designed for memory evaluation. Each environment simulates a 7-DoF arm with a two-finger gripper. Observations are paired RGB images ($3 \times 128 \times 128$) from a static and wrist camera; actions are continuous (7 joints + gripper). Rewards are binary, given only on task success. We study two families of MIKASA-Robo tasks: (i) `RememberColor[3,5,9]-v0`, where the agent must recall the color of a hidden cube after a delay with distractors, and (ii) `TakeItBack-v0`, where the agent first moves a cube to a goal, then must return it once the goal changes.

**POPGym benchmark.** POPGym benchmark (Morad et al., 2023a) is a large suite of 48 partially observable environments designed to stress agent memory. The tasks span two categories: (i) diagnostic memory puzzles, which require agents to remember cues or solve algorithmic sequence problems (e.g., copy, reverse, n-back, and long-horizon T-Mazes), and (ii) partially observable control tasks, which adapt classic control benchmarks such as CartPole, MountainCar, and LunarLander to observation-limited settings. This diversity allows POPGym to test both symbolic memory skills and generalization in continuous control, providing a broad and challenging benchmark for evaluating memory-augmented RL methods. Detailed results across all 48 POPGym tasks for each of considered baselines are presented in Table 5.

## A.5 TRAJECTORY CONDITIONING

To assess (1) how ELMUR behaves on standard high-dimensional MDP control tasks and (2) how different conditioning schemes influence performance, we evaluate it on the MuJoCo domains of the D4RL benchmark (Todorov et al., 2012). The datasets contain trajectories of heterogeneous quality, including many far from expert level, which typically requires conditioning on returns-to-go and actions (as in DT (Chen et al., 2021)) to enable trajectory stitching.

We therefore test two conditioning modes for ELMUR: $(R, o, a)$, analogous to DT, and $o$-only, as in standard behavior cloning. Despite the lower quality of D4RL data, ELMUR achieves an average normalized score of 76.4 under $(R, o, a)$ conditioning, only about 2% higher than with $o$-only conditioning. This demonstrates that the model remains robust to the choice of conditioning variables and does not rely on access to return-to-go or action labels to perform well.

Overall, ELMUR matches or exceeds the performance of several established offline RL baselines on these MDP tasks, confirming that its strong results are not restricted to memory-centric POMDP environments. Its competitive behavior across conditioning modes further highlights its versatility and ability to leverage suboptimal trajectories effectively.

Table 5: **POPGym benchmark results.** Reported scores are mean ± standard error, averaged over 3 runs and evaluated with 100 random seeds. Across 21 partially observable tasks spanning puzzle and control domains, ELMUR achieves the best performance on 12 tasks, underscoring the effectiveness of its memory module for reasoning under partial observability.

| POPGym-48 Task | RATE | DT | Random | BC-MLP | BC-LSTM | ELMUR | Expert PPO-GRU |
|---|---|---|---|---|---|---|---|
| AutoencodeEasy-v0 | -0.29±0.00 | -0.47±0.00 | -0.50±0.00 | -0.47±0.00 | -0.32±0.00 | -0.26±0.00 | -0.26 |
| AutoencodeMedium-v0 | -0.46±0.00 | -0.49±0.00 | -0.50±0.01 | -0.50±0.00 | -0.44±0.00 | -0.44±0.00 | -0.43 |
| AutoencodeHard-v0 | -0.47±0.00 | -0.49±0.00 | -0.50±0.00 | -0.49±0.00 | -0.47±0.00 | -0.46±0.00 | -0.48 |
| BattleshipEasy-v0 | -0.81±0.02 | -0.93±0.03 | -0.46±0.01 | -1.00±0.00 | -0.49±0.01 | -0.79±0.02 | -0.35 |
| BattleshipMedium-v0 | -0.92±0.01 | -0.97±0.01 | -0.41±0.00 | -1.00±0.00 | -0.67±0.01 | -0.79±0.01 | -0.40 |
| BattleshipHard-v0 | -0.91±0.02 | -0.91±0.03 | -0.39±0.01 | -1.00±0.00 | -0.81±0.02 | -0.80±0.00 | -0.43 |
| ConcentrationEasy-v0 | -0.06±0.02 | -0.05±0.01 | -0.19±0.01 | -0.92±0.00 | -0.14±0.00 | -0.14±0.01 | -0.12 |
| ConcentrationMedium-v0 | -0.25±0.00 | -0.25±0.01 | -0.19±0.00 | -0.92±0.00 | -0.19±0.01 | -0.24±0.00 | -0.44 |
| ConcentrationHard-v0 | -0.84±0.00 | -0.84±0.00 | -0.84±0.00 | -0.88±0.00 | -0.84±0.00 | -0.82±0.00 | -0.87 |
| CountRecallEasy-v0 | 0.07±0.01 | -0.46±0.01 | -0.93±0.00 | -0.92±0.00 | 0.05±0.00 | 0.03±0.01 | 0.22 |
| CountRecallMedium-v0 | -0.54±0.00 | -0.81±0.02 | -0.93±0.00 | -0.92±0.00 | -0.56±0.00 | -0.56±0.01 | -0.55 |
| CountRecallHard-v0 | -0.47±0.01 | -0.75±0.03 | -0.88±0.00 | -0.88±0.00 | -0.47±0.00 | -0.47±0.00 | -0.48 |
| HigherLowerEasy-v0 | 0.50±0.00 | 0.50±0.00 | 0.00±0.01 | 0.47±0.00 | 0.50±0.00 | 0.50±0.00 | 0.51 |
| HigherLowerMedium-v0 | 0.52±0.00 | 0.51±0.00 | 0.01±0.00 | 0.50±0.00 | 0.51±0.01 | 0.51±0.00 | 0.49 |
| HigherLowerHard-v0 | 0.50±0.00 | 0.50±0.00 | -0.01±0.00 | 0.49±0.00 | 0.50±0.00 | 0.50±0.00 | 0.49 |
| LabyrinthEscapeEasy-v0 | 0.95±0.00 | 0.80±0.01 | -0.39±0.01 | 0.72±0.05 | 0.92±0.01 | 0.92±0.00 | 0.95 |
| LabyrinthEscapeMedium-v0 | -0.56±0.01 | -0.67±0.04 | -0.84±0.04 | -0.71±0.03 | -0.69±0.02 | -0.54±0.01 | -0.49 |
| LabyrinthEscapeHard-v0 | -0.81±0.01 | -0.82±0.01 | -0.94±0.01 | -0.89±0.01 | -0.86±0.00 | -0.82±0.01 | -0.94 |
| LabyrinthExploreEasy-v0 | 0.95±0.00 | 0.88±0.06 | -0.34±0.01 | 0.87±0.01 | 0.93±0.00 | 0.96±0.00 | 0.96 |
| LabyrinthExploreMedium-v0 | 0.88±0.00 | 0.86±0.01 | -0.61±0.00 | 0.45±0.01 | 0.82±0.01 | 0.86±0.01 | 0.87 |
| LabyrinthExploreHard-v0 | 0.79±0.00 | 0.77±0.01 | -0.73±0.00 | 0.26±0.01 | 0.71±0.01 | 0.84±0.00 | 0.79 |
| MineSweeperEasy-v0 | 0.15±0.03 | -0.33±0.04 | -0.26±0.03 | -0.47±0.01 | 0.20±0.00 | -0.17±0.02 | 0.28 |
| MineSweeperMedium-v0 | -0.20±0.00 | -0.37±0.02 | -0.39±0.01 | -0.48±0.00 | -0.16±0.00 | -0.32±0.00 | -0.10 |
| MineSweeperHard-v0 | -0.44±0.00 | -0.40±0.01 | -0.43±0.00 | -0.49±0.00 | -0.35±0.01 | -0.39±0.01 | -0.27 |
| MultiarmedBanditEasy-v0 | 0.37±0.01 | 0.27±0.01 | 0.02±0.00 | 0.05±0.00 | 0.17±0.02 | 0.43±0.01 | 0.62 |
| MultiarmedBanditMedium-v0 | 0.32±0.01 | 0.35±0.01 | 0.01±0.00 | 0.21±0.01 | 0.14±0.00 | 0.32±0.01 | 0.59 |
| MultiarmedBanditHard-v0 | 0.22±0.03 | 0.27±0.01 | 0.01±0.00 | 0.01±0.00 | 0.17±0.01 | 0.22±0.01 | 0.43 |
| NoisyPositionOnlyCartPoleEasy-v0 | 0.88±0.03 | 0.87±0.02 | 0.11±0.00 | 0.23±0.00 | 0.44±0.01 | 0.59±0.02 | 0.98 |
| NoisyPositionOnlyCartPoleMedium-v0 | 0.33±0.01 | 0.34±0.00 | 0.12±0.01 | 0.18±0.00 | 0.25±0.01 | 0.27±0.00 | 0.57 |
| NoisyPositionOnlyCartPoleHard-v0 | 0.18±0.01 | 0.17±0.01 | 0.11±0.00 | 0.16±0.00 | 0.22±0.01 | 0.22±0.01 | 0.36 |
| NoisyPositionOnlyPendulumEasy-v0 | 0.87±0.00 | 0.84±0.01 | 0.27±0.01 | 0.31±0.00 | 0.88±0.00 | 0.88±0.00 | 0.90 |
| NoisyPositionOnlyPendulumMedium-v0 | 0.68±0.00 | 0.63±0.01 | 0.27±0.01 | 0.30±0.00 | 0.72±0.00 | 0.72±0.00 | 0.73 |
| NoisyPositionOnlyPendulumHard-v0 | 0.60±0.01 | 0.56±0.01 | 0.26±0.00 | 0.28±0.00 | 0.66±0.00 | 0.65±0.00 | 0.67 |
| PositionOnlyCartPoleEasy-v0 | 0.93±0.03 | 1.00±0.00 | 0.12±0.00 | 0.15±0.00 | 0.17±0.00 | 1.00±0.00 | 1.00 |
| PositionOnlyCartPoleMedium-v0 | 0.07±0.00 | 0.34±0.08 | 0.05±0.00 | 0.09±0.00 | 0.12±0.00 | 0.11±0.00 | 1.00 |
| PositionOnlyCartPoleHard-v0 | 0.05±0.01 | 0.03±0.00 | 0.04±0.00 | 0.05±0.00 | 0.06±0.00 | 0.10±0.00 | 1.00 |
| PositionOnlyPendulumEasy-v0 | 0.54±0.02 | 0.51±0.03 | 0.27±0.00 | 0.29±0.00 | 0.91±0.00 | 0.52±0.00 | 0.92 |
| PositionOnlyPendulumMedium-v0 | 0.49±0.01 | 0.55±0.01 | 0.26±0.00 | 0.30±0.00 | 0.89±0.00 | 0.50±0.01 | 0.88 |
| PositionOnlyPendulumHard-v0 | 0.47±0.01 | 0.49±0.01 | 0.26±0.00 | 0.28±0.00 | 0.82±0.00 | 0.63±0.01 | 0.82 |
| RepeatFirstEasy-v0 | 1.00±0.00 | 0.45±0.16 | -0.49±0.01 | -0.50±0.00 | 1.00±0.00 | 1.00±0.00 | 1.00 |
| RepeatFirstMedium-v0 | 0.99±0.01 | -0.21±0.18 | -0.50±0.00 | -0.50±0.00 | 0.99±0.01 | 0.99±0.01 | 1.00 |
| RepeatFirstHard-v0 | 0.10±0.02 | 0.42±0.14 | -0.50±0.00 | -0.50±0.00 | -0.50±0.00 | 0.99±0.01 | 0.99 |
| RepeatPreviousEasy-v0 | 1.00±0.00 | 1.00±0.00 | -0.49±0.01 | -0.52±0.00 | 1.00±0.00 | 1.00±0.00 | 1.00 |
| RepeatPreviousMedium-v0 | -0.38±0.01 | -0.38±0.00 | -0.50±0.01 | -0.50±0.00 | -0.38±0.00 | -0.39±0.00 | -0.39 |
| RepeatPreviousHard-v0 | -0.46±0.00 | -0.47±0.00 | -0.51±0.00 | -0.48±0.00 | -0.45±0.00 | -0.46±0.00 | -0.48 |
| VelocityOnlyCartpoleEasy-v0 | 1.00±0.00 | 1.00±0.00 | 0.11±0.00 | 0.99±0.00 | 1.00±0.00 | 1.00±0.00 | 1.00 |
| VelocityOnlyCartpoleMedium-v0 | 1.00±0.00 | 1.00±0.00 | 0.06±0.00 | 0.83±0.01 | 1.00±0.00 | 1.00±0.00 | 1.00 |
| VelocityOnlyCartpoleHard-v0 | 1.00±0.00 | 0.96±0.02 | 0.04±0.00 | 0.63±0.00 | 1.00±0.00 | 1.00±0.00 | 0.99 |
| **Sum of returns** | 9.54 | 5.80 | -12.24 | -6.83 | 8.96 | **10.41** | 16.51 |

## A.6 EXTENDED RELATED WORK

**Transformers for manipulation.** Work on transformer-based manipulation splits into characteristic families with complementary strengths and limits. *Perception-centric visuomotor transformers* emphasize 3D or multi-view geometry to map pixels to precise end-effector actions under (near) full observability, excelling at spatial alignment but assuming short temporal credit assignment: PerAct (Shridhar et al., 2023), RVT (Goyal et al., 2023), RVT-2 (Goyal et al., 2024). *Sequence/skill models* distill demonstrations into reusable action chunks and long-horizon trajectories, improving sample efficiency but remaining bottlenecked by finite context windows: ACT (Zhao et al., 2023), Skill Transformer (Huang et al., 2023), ILBiT (Kobayashi et al., 2025). *Planning/RL-driven transformers* blend sequence modeling with value learning or planning for closed-loop control under uncertainty, yet still inherit fixed-context limitations: OPTIMUS (Dalal et al., 2023), ActionFlow (Funk et al., 2024), Q-Transformer (Chebotar et al., 2023), CCT/ARP (Zhang et al., 2025b), FLaRe (Hu et al., 2025). *Alternative backbones* (state-space, diffusion) target long continuous horizons and smooth control but lack explicit persistent state: RoboMamba (Liu et al., 2024b), Diffusion Policy (Chi et al., 2023).

**VLA models for manipulation.**   VLA models broaden task coverage by conditioning on language and scaling data, while keeping the transformer core unchanged and context-bounded. *Instruction-following generalists* demonstrate broad skill repertoires with language prompts but no explicit long-term memory: RT-1 (Brohan et al., 2022), RT-2 (Zitkovich et al., 2023), Octo (Team et al., 2024), OpenVLA (Kim et al., 2024), VIMA (Jiang et al., 2022). *Efficiency/modularity variants* freeze large encoders and train lightweight adapters or experts for practicality at scale: RoboFlamingo (Li et al., 2023), CogACT (Li et al., 2024), FLOWER (Reuss et al., 2025), NinA (Tarasov et al., 2025). *Reasoning/hierarchy extensions* inject geometric priors, step-wise reasoning, or multi-level control while still relying on finite windows: 3D-VLA (Zhen et al., 2024), CoT-VLA (Zhao et al., 2025), TraceVLA (Zheng et al., 2024), HiRT (Zhang et al., 2024), DP-VLA (Han et al., 2024). *Specialized and hybrid designs* tailor the backbone to domain constraints (dexterous hands, spatial priors) or mix diffusion with autoregression: HybridVLA (Liu et al., 2025), DexVLA (Wen et al., 2025a), SpatialVLA (Qu et al., 2025), OpenVLA-OFT (Kim et al., 2025). *Generalist robot agents* pursue open-world embodiment with growing breadth but inherit the same temporal limitations: TinyVLA (Wen et al., 2025b), $\pi_0$ (Black et al., 2024), $\pi_{0.5}$ (Intelligence et al., 2025), GR00T N1 (Bjorck et al., 2025), Gemini Robotics (Team et al., 2025), NORA (Hung et al., 2025).

**Memory in Deep Learning.**   Mechanisms for long-term information fall into three broad tracks. *Implicit recurrence and long-range sequence models* retain information in hidden dynamics but offer limited, indirect control over storage and forgetting: LSTM (Hochreiter & Schmidhuber, 1997), xL-STM (Beck et al., 2024), linear-attention Transformers (Katharopoulos et al., 2020), RWKV (Peng et al., 2023), S4 (Gu et al., 2021), Mamba (Gu & Dao, 2023). *Explicit external memory with learned read–write* provides addressable storage with content-based access, trading off simplicity for optimization/scaling complexity: NTM (Graves et al., 2014), DNC (Graves et al., 2016), Memory Networks (Weston et al., 2014), differentiable memory for meta-learning (Santoro et al., 2016), recent variants (Ahmadi, 2020). *Transformer context extension* pushes horizons via cached activations, compression, or auxiliary memory modules but typically keeps memory peripheral to the core token computation: Transformer-XL (Dai et al., 2019), Compressive Transformer (Rae et al., 2019), Memorizing Transformer (Wu et al., 2022), Ring Attention (Liu et al., 2023), Memformer (Wu et al., 2020), associative-memory Transformers (Rodkin et al., 2024), ERNIE-style memory (Ding et al., 2020), test-time memorization (TiTANS) (Behrouz et al., 2024).

**Memory in RL.**   In partially observed decision processes, memory is not optional; it is the state estimator. *Spatial/episodic buffers* externalize salient facts in read–write maps or associative stores (navigational and episodic use-cases): Neural Map (Parisotto & Salakhutdinov, 2017), HCAM (Lampinen et al., 2021), Stable Hadamard Memory (Le et al., 2024). *Sequence-model adaptations* retrofit transformers for recurrence and stability in RL, improving long-horizon training but leaving persistence bounded by context: DTQN (Esslinger et al., 2022), GTrXL (Parisotto et al., 2020), AGaLiTe (Pramanik et al., 2023), AMAGO-2 (Grigsby et al., 2024). *Evaluation frameworks* formalize memory demands and failure modes under partial observability: (Cherepanov et al., 2026b; Yue et al., 2024; Wang et al., 2025b). *Transformers with external stores for RL* insert explicit memory alongside the policy, but often as a sequence-level attachment: RATE (Cherepanov et al., 2026c), FFM (Morad et al., 2023b), Re:Frame (Zelezetsky et al., 2025).

**Memory in manipulation tasks.**   Long-horizon, partially observed manipulation has motivated three practical extensions. *Trajectory summarization* compresses the past into a short token set, trading completeness for compactness: TraceVLA (Zheng et al., 2024). *External feature banks* cache recent visual features for retrieval, focusing attention but leaving distant history underrepresented: SAM2Act+ (Fang et al., 2025). *Structured memory modules* interface an explicit store with the policy to persist scene/task variables: MemoryVLA (Shi et al., 2025). *Hierarchies* shift temporal burden to slow planners or high-level controllers, which may mask but not remove the need for persistent state at the policy layer: HiRT (Zhang et al., 2024), DP-VLA (Han et al., 2024). For evaluation, *targeted benchmarks* isolate memory factors: MemoryBench (Fang et al., 2025); broader suites capture multiple memory types and delays: MIKASA-Robo (Cherepanov et al., 2026a).

Table 6: Returns on ViZDoom-Two-Colors (mean ± standard error over 6 runs). **Top-1** and **Top-2** results are highlighted.

| Method | Return (mean ± sem) |
|---|---|
| ELMUR | 55.28 ± 0.94 |
| RATE | 59.21 ± 1.19 |
| DT | 31.45 ± 1.21 |
| RMT | 53.53 ± 3.15 |
| TrXL | 49.62 ± 0.88 |
| BC-LSTM | 52.16 ± 2.59 |
| CQL-MLP | 12.49 ± 0.19 |
| Random | 4.78 |

## A.7 3D Visual Navigation

To evaluate ELMUR's ability to handle 3D long-horizon navigation with visual input, we additionally experiment in the ViZDoom-Two-Colors environment (Sorokin et al., 2022). The agent moves in a room with an acid floor that continuously drains health at every timestep. At the beginning of each episode it briefly observes a single colored pole (green or red) for 45 steps. The pole then disappears, and the agent must navigate among green and red objects scattered on the floor and collect only those whose color matches the initial pole in order to receive rewards and restore health. The results in Table 6 show that ELMUR matches the performance of the previous best baseline within statistical uncertainty, indicating that ELMUR is effective not only on memory-intensive manipulation tasks but also on challenging memory-dependent navigation tasks with 3D visual observations.

## A.8 Training Process

When training ELMUR, we compute the loss on every segment processed recursively, detaching the memory state between segments (i.e., without backpropagation through time). The choice of loss depends on the task domain: for T-Maze, CartPole-v1, and a subset of POPGym tasks with discrete actions, we apply a cross-entropy loss; for MIKASA-Robo and the remaining POPGym tasks with continuous actions, we use a mean-squared error (MSE) loss. The ELMUR hyperparameters used in the experiments are presented in The ELMUR hyperparameters used in the experiments are presented in Table 7.

For data, we follow a consistent offline imitation-learning setup. Each MIKASA-Robo environment provides a dataset of 1000 expert demonstrations generated by a PPO policy trained with oracle-level state access. For T-Maze, we collect 6000 successful oracle-level trajectories. For POPGym, we adopt the datasets of Morad et al. (2023a), consisting of 3000 trajectories per environment generated by a PPO-GRU expert. Finally, for CartPole-v1, we use 1000 successful trajectories collected from a pre-trained PPO policy.

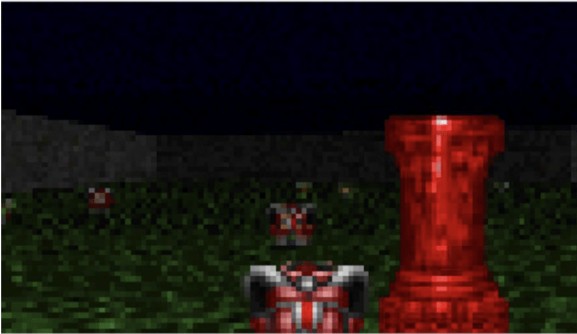

Figure 8: **ViZDoom-Two-Colors environment.**

| Parameter | RememberColor | TakeItBack | T-Maze | POPGym-AutoencodeEasy | CartPole-v1 |
|---|---|---|---|---|---|
| $d_{\text{model}}$ | 128 | 32 | 128 | 64 | 128 |
| Routed $d_{\text{ff}}$ | 128 | 256 | 32 | 128 | 128 |
| Shared $d_{\text{ff}}$ | 128 | 32 | 512 | 256 | 256 |
| Layers | 4 | 2 | 2 | 12 | 4 |
| Heads | 16 | 16 | 2 | 4 | 4 |
| Experts (MoE) | 16 | 1 | 2 | 1 | 4 |
| Shared Experts | 1 | 1 | 2 | 2 | 1 |
| Top-$k$ routing | 2 | 1 | 3 | 1 | 2 |
| Memory size | 256 | 32 | 2 | 8 | 16 |
| Memory init std | 0.1 | 0.001 | 0.001 | 0 | 0.01 |
| Memory dropout | 0.06 | 0.23 | 0.01 | 0.17 | 0.05 |
| LRU blend $\alpha$ | 0.40 | 0.20 | 0.05 | 0.80 | 0.9 |
| Dropout | 0.13 | 0.01 | 0.10 | 0.14 | 0.10 |
| Dropatt | 0.30 | 0.18 | 0.17 | 0.26 | 0.10 |
| Label smoothing | 0.21 | 0.20 | 0.16 | 0.22 | 0.00 |
| Context length | 20 | 60 | 10 | 35 | 30 |
| Batch size | 64 | 64 | 128 | 128 | 512 |
| Learning rate | 2.05e-4 | 2.57e-4 | 2.06e-4 | 1.16e-4 | 3.0e-4 |
| Warmup steps | 30000 | 30000 | 10000 | 50000 | 1000 |
| Cosine decay | True | False | True | False | True |
| LR end factor | 0.1 | 0.01 | 1.0 | 0.01 | 0.1 |
| Weight decay | 0.001 | 0.01 | 0.0001 | 0.1 | 0.01 |
| Epochs | 200 | 300 | 1000 | 800 | 100 |
| Grad clip | 5 | 1 | 5 | 5 | 1.0 |
| Beta1 | 0.99 | 0.9 | 0.95 | 0.99 | 0.9 |
| Beta2 | 0.99 | 0.99 | 0.999 | 0.99 | 0.999 |

Table 7: **Hyperparameters for ELMUR.** We report all architecture and training parameters used across tasks.

## A.9 MEMORY PROBING

To examine whether ELMUR stores task-relevant information in its external memory rather than relying solely on increased parameter count, we perform a controlled memory probing study on the `RememberColor3-v0` environment. The goal is to determine whether memory embeddings contain linearly or nonlinearly decodable information about the target cube color, which serves as the minimal sufficient statistic required for correct decision making in the recall stage of the task.

**Dataset construction.** We first train an ELMUR policy to expert-level performance on `RememberColor3-v0`. Using the trained policy, we collect 1000 evaluation episodes and extract, at every timestep, all layer-local memory embeddings and the corresponding ground-truth target color. We retain only episodes that successfully complete the task to avoid confounding effects from incorrect trajectories and to ensure that the probed representations correspond to functional memory usage. The resulting dataset contains tuples of the form

$$(\ell, t, m_t^{(\ell)}, y),$$

where $\ell$ is the layer index, $t$ is the timestep at which the memory vector was recorded, $m_t^{(\ell)} \in \mathbb{R}^{M \times d}$ is the memory embedding of layer $\ell$ at timestep $t$, and $y \in \{0, 1, 2\}$ is the target color label.

**Probing protocol.** Memory in ELMUR is structured and evolves differently across layers and timesteps. To distinguish these effects, we construct a separate probing dataset for every layer $\ell \in \{0, \dots, L-1\}$ and every chosen timestep $t$ (we use $t \in \{0, 1, 5, 10, 20\}$). For each pair $(\ell, t)$ we train an independent probe on the corresponding memory vectors $\{m_t^{(\ell)}\}$.

Each probe is a three-layer multilayer perceptron (MLP) with hidden size 512 and ReLU nonlinearities. We train the probes with a cross-entropy loss to predict the target cube color solely from the memory vector. Probes are trained with Adam for 200 epochs using an 80/20 train-validation split. Importantly, probes are not allowed to influence the base policy; the probing procedure is strictly post-hoc.

**Memory Evaluation.** For every $(\ell, t)$ pair we report the accuracy and confusion matrix of the corresponding probe. At $t=0$, when memory is initialized with random noise, all layers yield roughly

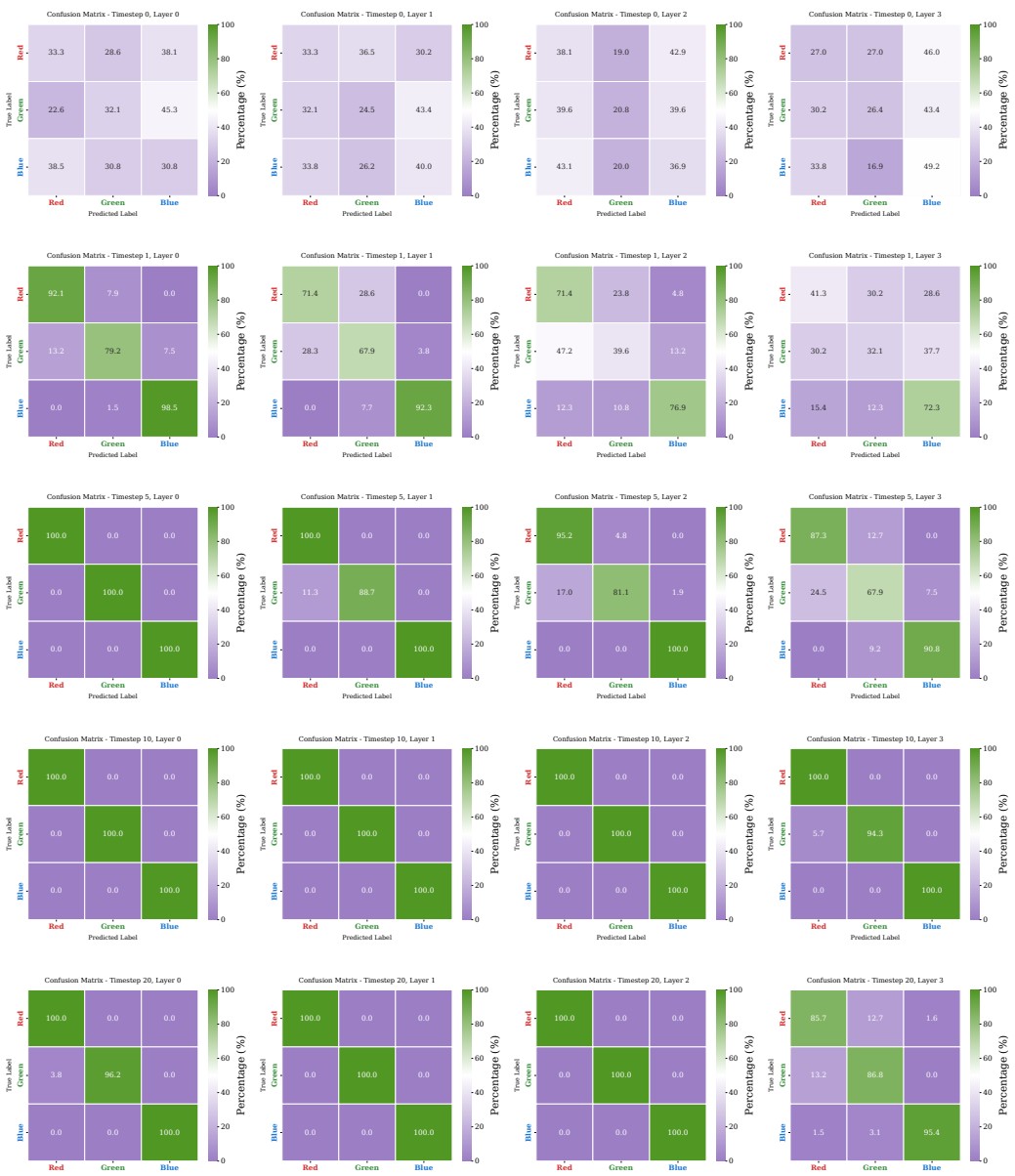

Figure 9: **Memory probing confusion matrices on `RememberColor3-v0`.** Each panel shows the confusion matrix (percentage) of the color probe trained on ELMUR memory embeddings for a specific timestep (rows, $t \in \{0, 1, 5, 10, 20\}$) and layer (columns, $\ell \in \{0, 1, 2, 3\}$). At $t{=}0$ (top row), when memory is still random, all probes sit near the 33% chance level, indicating that the target color is not decodable from initialization. After the first write at $t{=}1$, the first layer already supports accurate decoding, with performance gradually decreasing for deeper layers. At $t{=}5$, once the cue has disappeared, probes achieve almost perfect accuracy, including 100% for the first layer. At the decision time $t{=}10$ and late in the episode at $t{=}20$, prediction remains essentially perfect in the first three layers and very high in the last layer, demonstrating stable long-horizon retention of the target color in memory.

33% accuracy per class, indicating that the target color is not decodable from the initial memory embeddings. After the first update at $t{=}1$, the first layer already supports reliable decoding for all three colors, with accuracy gradually decreasing for deeper layers, which shows that the model writes a usable color code into memory immediately after observing the cue. By $t{=}5$, when the colored cube

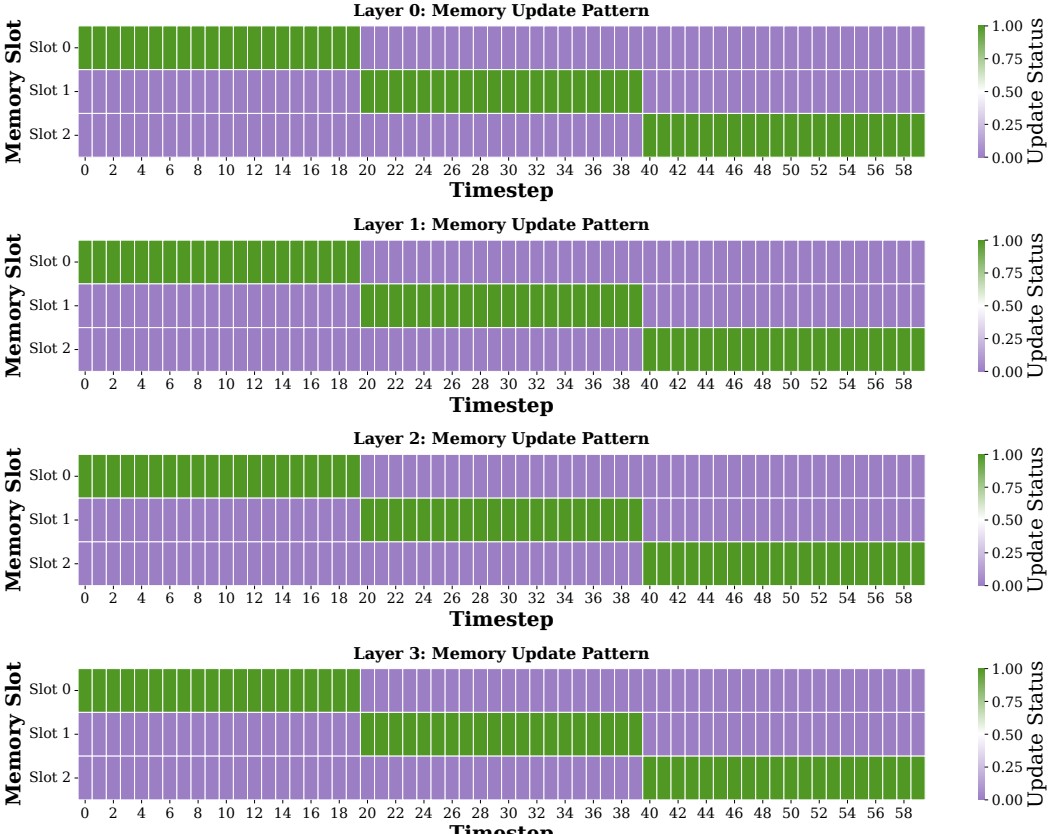

Figure 10: **Memory update patterns across layers on `RememberColor3-v0`.** Each heatmap shows, for one ELMUR layer, which memory slot is updated at each timestep (horizontal axis: timestep $t \in [0, 60]$, vertical axis: slots 0–2; green indicates an update, purple indicates no update). All layers exhibit the same structured schedule: early in the episode only slot 0 is updated, then responsibility shifts to slot 1, and finally to slot 2, producing a clean rotation of writes enforced by the LRU policy. After a slot becomes inactive it is not modified again, which implies that once the target color has been written into a slot its content is preserved for the remainder of the episode without further overwriting.

has disappeared from the scene, prediction accuracy improves further, reaching $100\%$ for the first layer and remaining high for the others. At $t=10$, when the agent sees all three cubes and must select the target, the probes achieve $100\%$ accuracy on the first three layers and near perfect accuracy on the last layer, and the same pattern persists at $t=20$. Aggregated confusion matrices across layers and timesteps are shown in Figure 9.

Together, these findings confirm that ELMUR allocates memory capacity to storing the latent task variable required to solve the environment and that its superior performance arises from functional memory usage rather than increased model size. In addition, the memory update patterns in Figure 10 show a consistent and interpretable write schedule across layers: each episode produces a single dominant write into a dedicated slot immediately after observing the cue, followed by long spans of preservation with no overwriting. The LRU mechanism activates only when necessary, producing sparse, well-timed updates rather than continuous churn.

The inter-slot similarity analysis in Figure 11 further supports this behavior. At the moment when the target cube is first observed, all layers exhibit a sharp transition in cosine similarity structure: a single slot becomes highly similar across repeated episodes while the remaining slots remain decorrelated. This indicates that ELMUR allocates one stable slot as the canonical container for the task variable. After this write event, the similarity trajectories remain nearly constant throughout

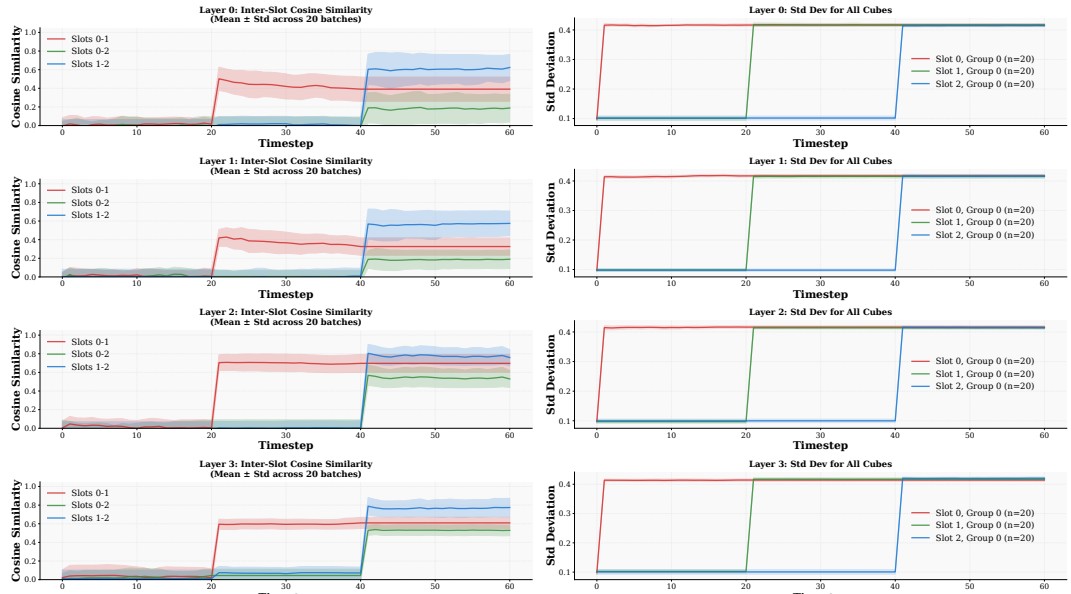

Figure 11: **Inter–slot similarity and variance of memory embeddings on `RememberColor3-v0`.** For each ELMUR layer (rows), the left panels show the cosine similarity between pairs of memory slots (0–1, 0–2, 1–2) as a function of timestep, averaged over 20 batches (shaded region: standard deviation). Around the transition to the second segment ($t = 20$) the similarities jump from near zero to higher, stable values, indicating that one slot acquires a structured representation which becomes partially shared with the others. A second transition to the third segment appears at $t = 40$, when another slot is written, after which the similarity profiles remain nearly constant, consistent with persistent storage. The right panels show the standard deviation of embeddings within each slot over episodes. Only the currently written slot exhibits a sharp increase in variance at its corresponding write time, while the other slots remain nearly constant, confirming that the model performs localized, slot-specific writes followed by stable retention of the encoded content.

the blank stage and the distractor stage, demonstrating that the stored representation is preserved without drift.

The slotwise variance plots show the same phenomenon from a complementary angle. The chosen slot exhibits a sudden rise in variance immediately after the cue is written, reflecting the encoding of task-specific information, while the unused slots retain near-zero variance for the remainder of the episode. Throughout the delay period the variance of the active slot remains stable, indicating that the stored color representation is neither degraded nor rewritten.

Taken together, the update patterns, similarity dynamics, and variance profiles reveal a coherent mechanism: ELMUR performs a targeted, one-shot write into a designated memory slot, then maintains this representation with high stability until retrieval, even in the presence of visually similar distractors. This behavior aligns precisely with the probing results and provides strong causal evidence that ELMUR employs explicit long-term memory rather than incidental capacity effects. A more detailed, layer-wise visualization of evolution of memory embeddings across the entire episode is presented in Figure 12.

## A.10 PCA MEMORY ANALYSIS

The geometric structure of memory representations is further illustrated by the Principal Component Analysis (PCA; Abdi & Williams (2010)) in Figure 13. The top row shows projections colored by memory slot (slots 0, 1, and 2), while the bottom row shows projections colored by layer. From left to right, we display all target colors together, then restrict to red, green, and blue targets respectively. In the slot based views, embeddings from different slots largely occupy the same manifolds for a

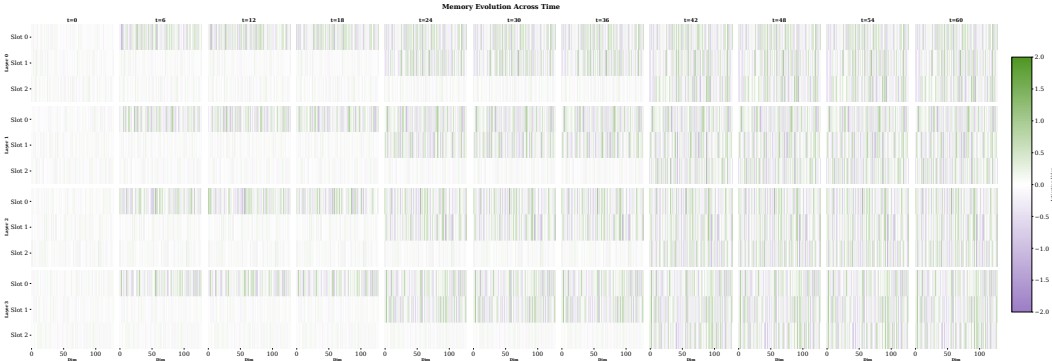

Figure 12: **Memory evolution across layers, slots, and time on `RememberColor3-v0`.** Each block corresponds to a snapshot of the memory state at a given timestep (columns, $t \in \{0, 6, \ldots, 60\}$). Within each block, rows index memory slots (0–2) for a given layer (stacked vertically from Layer 0 to Layer 3), and the horizontal axis shows embedding dimensions. Color encodes activation value. Early timesteps exhibit near-random low-amplitude activations, while after the cue is observed a single slot in each layer becomes strongly structured and remains stable over subsequent timesteps. Later in the episode additional slots are written once and then preserved, illustrating that ELMUR performs sparse, slot-specific updates followed by long-horizon retention of the encoded representation.

fixed color, indicating that slots are not tied to particular colors and instead share a common content space. In the layer based views, points from different layers form well separated clusters, especially when conditioning on a single color, which suggests that deeper layers produce progressively more structured and compact encodings of the target color while maintaining clear separation between colors in the joint view.

The temporal evolution of memory representations is illustrated in Figure 14, which shows PCA projections of all memory vectors with color indicating timestep. The first panel aggregates all target colors, while the remaining three panels condition on red, green, and blue targets respectively. In all cases the trajectories exhibit a characteristic pattern: memories start from a diffuse cloud near the initialization region (early timesteps in purple), then rapidly converge onto a small number of low-dimensional manifolds after the cue is written, and remain confined there for the rest of the episode (late timesteps in green). Conditioning on a single color reveals that these manifolds are largely color specific, with each target color occupying its own stable region in PCA space and showing minimal drift over time. This behavior indicates that ELMUR quickly encodes the target color into a compact representation and maintains it throughout the delay and retrieval phases without substantial degradation.

## A.11 CROSS-ATTENTION MAPS

Figure 15 visualizes the cross-attention patterns of the write (`tok2mem`, left) and read (`mem2tok`, right) modules. Each row corresponds to one of the 16 attention heads, the horizontal axis shows timesteps (0-60), and the vertical axis indexes the three memory slots. In the `tok2mem` maps, attention is almost entirely suppressed at the beginning of the episode and then exhibits a short, high-intensity burst around the cue presentation, concentrated on a single slot across many heads. After this write event, activity becomes sparse with only occasional weak updates, consistent with a one-shot allocation of the cue into a dedicated slot followed by long-term preservation. In contrast, the `mem2tok` maps display more sustained but still structured activity: once the cue has been written, several heads repeatedly attend to the same slot throughout the blank interval and the decision phase, with peaks near the timesteps where the agent must choose the correct cube. Other heads remain nearly inactive or attend to alternative slots, suggesting head specialization. Overall, these attention patterns corroborate the memory probing and similarity analyses, showing that ELMUR performs a sharp, localized write of the target color into a single slot and then repeatedly reads from that slot during subsequent reasoning and action.

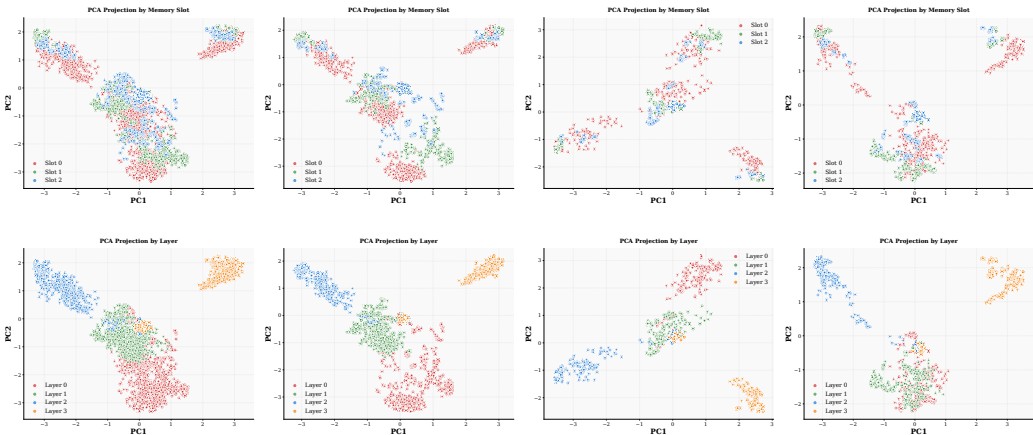

Figure 13: **PCA structure of memory embeddings by slot and layer on `RememberColor3-v0`.** Each panel shows a 2D PCA projection of all memory vectors. The top row colors points by memory slot (0, 1, 2), while the bottom row colors them by layer (0, 1, 2, 3). From left to right, we display embeddings for all target colors, and then restricted to red, green, and blue targets respectively. In the slot-based views, embeddings from different slots largely occupy the same manifolds for a fixed color, indicating that slots share a common content space rather than being tied to particular colors. In contrast, the layer-based views form well-separated clusters, especially when conditioning on a single color, showing that deeper layers produce progressively more specialized and compact representations of the same latent task variable.

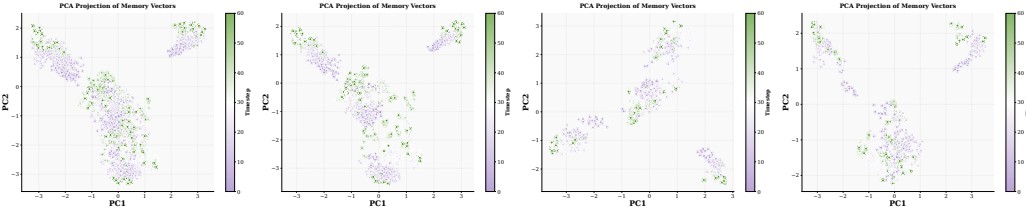

Figure 14: **PCA projection of memory embeddings over time.** Each panel shows memory vectors from all layers and slots in `RememberColor3-v0`, projected onto the first two principal components and colored by timestep (see colorbar). From left to right we plot all episodes, and then subsets conditioned on the target cube being red, green, or blue. Early timesteps occupy a diffuse region of the space, while vectors shortly after the cue observation move into compact, color-specific clusters that remain stable for the rest of the episode, indicating that the memory state rapidly converges to a persistent representation of the target color.

## A.12 LIMITATIONS

ELMUR uses a simple LRU rule with fixed blending, which makes its memory mechanism transparent and easy to analyze, though future work could explore adaptive variants. Segment-level recurrence adds a small cross-attention cost per layer, but this cost scales with the fixed number of memory slots rather than sequence length, making efficiency predictable even in long horizons. MoE-based FFNs already provide parameter efficiency at our scale, and larger architectures may further amplify this benefit. Finally, our study focuses on synthetic, POPGym, and simulated robotic tasks under IL, giving controlled and reproducible insights; extending to online RL and real-robot deployments offers promising next steps.

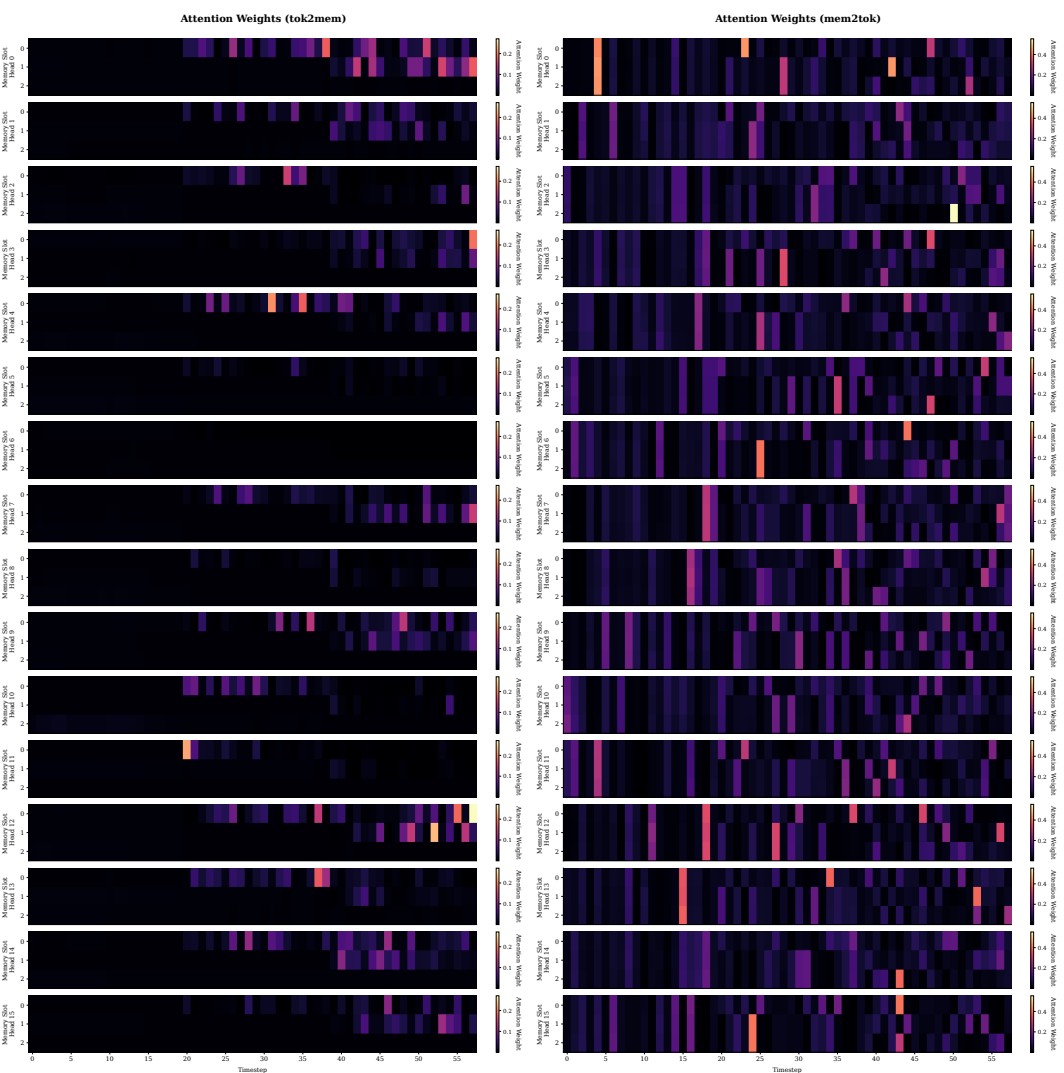

Figure 15: **Cross–attention patterns between tokens and memory slots in ELMUR.** Heatmaps show attention weights for the *tok2mem* module (left) and the *mem2tok* module (right) in the RememberColor3-v0 task. Each row corresponds to one of the 16 attention heads, the horizontal axis is the timestep ($t \in [0, 60]$), and the vertical axis indexes the three memory slots. Color intensity indicates the magnitude of the attention weight. tok2mem heads exhibit sparse, sharply localized write events into a single slot around cue and update timesteps, whereas mem2tok heads display broader read patterns that concentrate on the slot carrying the target color during the decision phase, consistent with targeted write, once storage followed by repeated retrieval of the latent task variable.

Table 8: **Offline RL baselines on the MIKASA-Robo benchmark.** We compare transformer-based methods (RATE, DT), behavior cloning (BC), classical offline RL algorithms (CQL), and Diffusion Policy (DP) across all 32 tasks. Scores are reported as mean ± sem over three seeds, with each seed averaged over 100 evaluation episodes. All models are trained from RGB observations (top-view and wrist-view cameras) under sparse, success-only rewards. Even with access to memory, existing baselines (RATE, DT) fail on most tasks, whereas ELMUR achieves the highest performance on 21 of the 23 tasks with non zero success rates and exceeds the previous state-of-the-art, RATE, by roughly 70% in aggregate success across all 32 tasks.

| Environment | RATE | DT | BC | CQL | DP | ELMUR |
|---|---|---|---|---|---|---|
| ShellGameTouch-v0 | 0.92±0.01 | 0.53±0.07 | 0.28±0.01 | 0.16±0.04 | 0.18±0.02 | 0.96±0.02 |
| ShellGamePush-v0 | 0.78±0.06 | 0.62±0.14 | 0.27±0.01 | 0.25±0.01 | 0.22±0.03 | 0.87±0.04 |
| ShellGamePick-v0 | 0.02±0.01 | 0.00±0.00 | 0.01±0.01 | 0.00±0.00 | 0.01±0.00 | 0.66±0.01 |
| InterceptSlow-v0 | 0.23±0.02 | 0.40±0.02 | 0.37±0.06 | 0.25±0.01 | 0.33±0.05 | 0.61±0.04 |
| InterceptMedium-v0 | 0.32±0.02 | 0.56±0.01 | 0.31±0.14 | 0.03±0.01 | 0.68±0.02 | 0.64±0.03 |
| InterceptFast-v0 | 0.30±0.04 | 0.36±0.04 | 0.03±0.02 | 0.02±0.02 | 0.21±0.05 | 0.54±0.02 |
| InterceptGrabSlow-v0 | 0.09±0.03 | 0.00±0.00 | 0.28±0.18 | 0.03±0.00 | 0.03±0.01 | 0.08±0.03 |
| InterceptGrabMedium-v0 | 0.09±0.03 | 0.00±0.00 | 0.11±0.02 | 0.08±0.04 | 0.03±0.01 | 0.10±0.04 |
| InterceptGrabFast-v0 | 0.14±0.03 | 0.11±0.03 | 0.09±0.02 | 0.08±0.03 | 0.18±0.02 | 0.20±0.01 |
| RotateLenientPos-v0 | 0.11±0.04 | 0.01±0.01 | 0.15±0.03 | 0.16±0.02 | 0.11±0.02 | 0.24±0.02 |
| RotateLenientPosNeg-v0 | 0.29±0.03 | 0.05±0.02 | 0.22±0.01 | 0.12±0.02 | 0.14±0.05 | 0.26±0.00 |
| RotateStrictPos-v0 | 0.03±0.02 | 0.05±0.04 | 0.01±0.00 | 0.03±0.01 | 0.06±0.02 | 0.16±0.00 |
| RotateStrictPosNeg-v0 | 0.08±0.01 | 0.05±0.03 | 0.04±0.02 | 0.04±0.02 | 0.15±0.01 | 0.12±0.02 |
| TakeItBack-v0 | 0.42±0.24 | 0.08±0.04 | 0.33±0.10 | 0.04±0.01 | 0.05±0.02 | 0.78±0.03 |
| RememberColor3-v0 | 0.65±0.04 | 0.01±0.01 | 0.27±0.03 | 0.29±0.01 | 0.32±0.01 | 0.89±0.07 |
| RememberColor5-v0 | 0.13±0.03 | 0.07±0.05 | 0.12±0.01 | 0.15±0.02 | 0.10±0.02 | 0.19±0.03 |
| RememberColor9-v0 | 0.09±0.02 | 0.01±0.01 | 0.12±0.02 | 0.15±0.01 | 0.17±0.01 | 0.23±0.02 |
| RememberShape3-v0 | 0.21±0.04 | 0.05±0.04 | 0.31±0.04 | 0.20±0.10 | 0.32±0.05 | 0.76±0.01 |
| RememberShape5-v0 | 0.17±0.04 | 0.04±0.04 | 0.18±0.01 | 0.15±0.00 | 0.21±0.04 | 0.27±0.02 |
| RememberShape9-v0 | 0.05±0.00 | 0.05±0.02 | 0.10±0.02 | 0.14±0.01 | 0.11±0.02 | 0.17±0.03 |
| RememberShapeAndColor3x2-v0 | 0.14±0.02 | 0.04±0.02 | 0.13±0.02 | 0.11±0.05 | 0.14±0.02 | 0.19±0.02 |
| RememberShapeAndColor3x3-v0 | 0.08±0.03 | 0.06±0.06 | 0.09±0.02 | 0.09±0.02 | 0.16±0.01 | 0.19±0.02 |
| RememberShapeAndColor5x3-v0 | 0.07±0.02 | 0.01±0.01 | 0.09±0.01 | 0.09±0.02 | 0.11±0.03 | 0.13±0.01 |
| BunchOfColors3-v0 | 0.00±0.00 | 0.00±0.00 | 0.00±0.00 | 0.00±0.00 | 0.00±0.00 | 0.00±0.00 |
| BunchOfColors5-v0 | 0.00±0.00 | 0.00±0.00 | 0.00±0.00 | 0.00±0.00 | 0.00±0.00 | 0.00±0.00 |
| BunchOfColors7-v0 | 0.00±0.00 | 0.00±0.00 | 0.00±0.00 | 0.00±0.00 | 0.00±0.00 | 0.00±0.00 |
| SeqOfColors3-v0 | 0.00±0.00 | 0.00±0.00 | 0.00±0.00 | 0.00±0.00 | 0.00±0.00 | 0.00±0.00 |
| SeqOfColors5-v0 | 0.00±0.00 | 0.00±0.00 | 0.00±0.00 | 0.00±0.00 | 0.00±0.00 | 0.00±0.00 |
| SeqOfColors7-v0 | 0.00±0.00 | 0.00±0.00 | 0.00±0.00 | 0.00±0.00 | 0.00±0.00 | 0.00±0.00 |
| ChainOfColors3-v0 | 0.00±0.00 | 0.00±0.00 | 0.00±0.00 | 0.00±0.00 | 0.00±0.00 | 0.00±0.00 |
| ChainOfColors5-v0 | 0.00±0.00 | 0.00±0.00 | 0.00±0.00 | 0.00±0.00 | 0.00±0.00 | 0.00±0.00 |
| ChainOfColors7-v0 | 0.00±0.00 | 0.00±0.00 | 0.00±0.00 | 0.00±0.00 | 0.00±0.00 | 0.00±0.00 |
| **Sum Across All Tasks** | 5.42 | 3.15 | 3.92 | 2.66 | 4.02 | **9.24** |

