# OpenReview forum: "ELMUR: External Layer Memory with Update/Rewrite for Long-Horizon RL Problems"
_ICLR.cc/2026/Conference — ICLR 2026 Poster_

### Official Review · Reviewer_AzD4 · 2025-10-30

**Soundness:** 3
**Presentation:** 3
**Contribution:** 2
**Rating:** 6
**Confidence:** 3

**Summary:**

ELMUR (External Layer Memory with Update/Rewrite) is a Transformer architecture that introduces structured, layer-wise external memory. Each layer maintains and updates memory embeddings through bidirectional cross-attention and an LRU-based update mechanism, enabling extremely long-term dependency modeling: up to 100,000× beyond the standard attention window. ELMUR achieves superior performance on long-horizon, partially observable tasks such as T-Maze, POPGym, and MIKASA-Robo, demonstrating a simple yet scalable solution for decision-making with long-range dependencies.

**Strengths:**

This paper proposes a new ELMUR mechanism and demonstrates its strong generalization ability through experiments.

**Weaknesses:**

1. **Table Placement and Layout:**

Table 1 is placed too low and is visually disconnected from the surrounding text. It would be better to reposition the table for improved readability and consistency. The overall layout of the paper requires further optimization.

2. **Ablation Study and Update Formula:**

The ablation study shows that the parameter λ is unstable when M<N. This instability might stem from the blending update formula. The motivation for choosing the specific update rule in Eq.8 should be explained in greater depth.

3. **Uniform Update Weight Across Segments:**

The paper applies the same update weight to all segments, which seems somewhat unreasonable. In practice, different segments may have varying importance — for example, key frames might require longer retention (i.e., a smaller λ). This issue is not discussed in the current analysis or in the future work section.

4. **Applicability and Generalization:**

How generalizable is this improvement? Is it limited to the Transformer architecture used in this paper’s experiments? The authors should apply ELMUR to more model architectures to demonstrate its generality or at least analyze its applicable scenarios.

5. **Bidirectional Update Visualization:**

The bidirectional update mechanism would benefit from a visual analysis. For instance, when processing a specific frame’s token, is the most relevant portion of memory indeed assigned the highest weight? Otherwise, the observed performance gain might simply result from adding a new learnable external module.

6. **Comparison with Prior Memory Methods:**

A comparison table should be provided to clearly illustrate the differences between ELMUR and prior memory-related approaches.

**Questions:**

1. What is the core motivation behind selecting the specific update rule presented in Eq. 8?

2. Why not consider adaptive strategies that adjust the update behavior based on the relative importance of different segments?

3. How generalizable is the observed improvement across different architectures？

4. Why not include a visual analysis illustrating the proposed bidirectional update mechanism?

---

> ### Author Response · Authors · 2025-11-25
> **Official Comment by Authors (part 1/2)**
>
> We thank Reviewer AzD4 for the thoughtful and constructive feedback. We particularly appreciate the recognition of ELMUR’s strong generalization ability and the clarity of its presentation, as well as the acknowledgment that the proposed mechanism offers a simple and scalable solution for long-horizon decision making.
>
> > W1. Table Placement and Layout: Table 1 is placed too low and is visually disconnected from the surrounding text. It would be better to reposition the table for improved readability and consistency. The overall layout of the paper requires further optimization.
>
> Thank you for the suggestion. We've improved the paper layout in the updated version of this paper. Please see the attached .pdf file.
>
> > W2, Q1. Ablation Study and Update Formula: The ablation study shows that the parameter λ is unstable when M<N. This instability might stem from the blending update formula. The motivation for choosing the specific update rule in Eq.8 should be explained in greater depth.
>
> The regime $M < N$ is indeed unstable, requiring additional tuning of a single hyperparameter $\lambda$, and the convex blending update in Eq. 8 was chosen for specific structural reasons.
> First, it yields a simple and analytically tractable forgetting mechanism: repeated updates to the same slot reduce to an exponential moving average of past writes, which allows closed form characterization of retention behavior and half life. This tractability relies on the linear, convex form of Eq. 8 and would not hold under more complex, nonlinear gating rules.
>
> Second, the convex update guarantees boundedness: as long as incoming writes are norm bounded, each memory vector remains within a fixed-radius ball. This uniform stability is critical for long trajectories in robotics, where more expressive additive updates could accumulate unbounded drift.
>
> Third, the sharp performance drop in the $M < N$ regime is a capacity bottleneck rather than a defect of the update rule. When the number of distinct segments that must be represented exceeds the available slots, any update rule that actually modifies memory must trade off stability and plasticity. Our ablations show that once $M \ge N$, the same update mechanism becomes stable and achieves near-perfect performance.
>
> We expanded Section 3 and the ablation discussion to clarify these points and to explicitly motivate the choice of Eq. 8 as the simplest stable rule that offers a tunable stability-plasticity tradeoff through $\lambda$, admits rigorous analysis, and behaves robustly in the well-provisioned regime.
>
> > W3, Q2. Uniform Update Weight Across Segments: The paper applies the same update weight to all segments, which seems somewhat unreasonable. In practice, different segments may have varying importance — for example, key frames might require longer retention (i.e., a smaller λ). This issue is not discussed in the current analysis or in the future work section.
>
> We keep $\lambda$ fixed for several reasons. First, a uniform update weight induces stable, time-homogeneous memory dynamics and defines the contraction behavior of the LRU overwrite step, which is necessary for the retention and boundedness analysis presented in Section 4. Allowing $\lambda$ to depend on the state or on the episode segment would make the update operator nonstationary and would break these guarantees.
>
> Second, our default training configuration detaches the memory pathway (see L178) to ensure stable optimization and reduce GPU memory consumption. Under this setting, gradients do not propagate through memory and therefore cannot reach $\lambda$, which makes learning it infeasible without reintroducing full gradient flow.
>
> To address the Reviewer's question directly, we conducted an additional experiment on RememberColor3-v0 in which $\lambda$ was made learnable and the memory pathway was no longer detached. The success rate remained similar to the fixed-$\lambda$ configuration, but GPU RAM usage approximately doubled due to the need to track gradients through all memory updates. This confirms that a fixed $\lambda$ is both computationally advantageous and empirically sufficient.
>
> We also note that even with a constant $\lambda$, semantic variation across segments is handled implicitly by the tok2mem and mem2tok cross-attention mechanisms: segments carrying higher information content naturally induce larger effective updates under a fixed update weight. Introducing an explicit content-adaptive or state-dependent update weight is an interesting direction for future work and would require new stability analysis. We appreciate the Reviewer's suggestion and agree it represents a promising avenue for further exploration.

---

> ### Author Response · Authors · 2025-11-25
> **Official Comment by Authors (part 2/2)**
>
> > W4, Q3. Applicability and Generalization: How generalizable is this improvement? Is it limited to the Transformer architecture used in this paper’s experiments? The authors should apply ELMUR to more model architectures to demonstrate its generality or at least analyze its applicable scenarios.
>
> ELMUR is introduced as a transformer architecture, not as a universal memory module intended to be attached to arbitrary sequence models. Its design relies on specific structural features of transformers, in particular the separation between self-attention and cross-attention, and the segment-level recurrence interface. The bidirectional token-memory interactions and the LRU-managed external slots are integrated at the layer level in a way that is tightly coupled to this architecture.
>
> For this reason, we do not view ELMUR as a drop-in mechanism that should be applied to unrelated model families such as state-space models or RNNs. Instead, the goal is to provide a principled transformer variant with stable, bounded long-term memory suitable for long-horizon decision making.
>
> Extending ELMUR-style external memory to fundamentally different architectures would require redesigning the read-write pathways and deriving new stability guarantees. We see this as an interesting direction for future work, but beyond the current paper’s scope.
>
> > W5, Q4. Bidirectional Update Visualization: The bidirectional update mechanism would benefit from a visual analysis. For instance, when processing a specific frame’s token, is the most relevant portion of memory indeed assigned the highest weight? Otherwise, the observed performance gain might simply result from adding a new learnable external module.
>
> To address the Reviewer’s request directly, we included attention visualizations in the revised text (see .pdf, Appendix A8, A9, A10 and Figures 9, 10, 11, 12, 13, 14, 15), which show that tokens assign the highest weights to memory slots containing temporally or semantically relevant information.
>
> > W6. Comparison with Prior Memory Methods: A comparison table should be provided to clearly illustrate the differences between ELMUR and prior memory-related approaches.
>
> Thank you for the suggestion. Summarizing the design differences between ELMUR and prior memory architectures helps clarify its contribution. In the revised .pdf version, we added a comparison table in the Appendix that highlights the key distinctions: (i) ELMUR uses layer-local external memory rather than global slots or cached activations, (ii) memory updates rely on explicit bidirectional cross-attention rather than concatenation or implicit recurrence, and (iii) bounded retention is maintained through an LRU-based overwrite rule with a tunable stability-plasticity tradeoff.
>
> Please **take a look at the revised version of the paper.** We improved the layout of the figures and tables in the main text; ran experiments on all 32 MIKASA-Robo tasks, achieving the best success rate on 21 out of 23 tasks and improving the aggregate success rate across all tasks by about 70% over the previous best baseline (see Appendix, Figure 8); added experiments on the D4RL MuJoCo benchmark with MDP tasks (see Appendix A.5, Table 4); added experiments on 3D visual navigation (see Appendix A.7, Table 6); added memory probing experiments to study what is stored in memory (see Appendix A.9, Figure 9); and added visualizations and analysis of the memory update process, including memory dynamics and attention maps (see Appendix A.10–A.11, Figures 10–15). All changes in the text are highlighted in blue.
>
>
> We thank Reviewer AzD4 for the thoughtful and constructive comments. We believe the clarifications and revisions provided above adequately address the raised concerns, and we welcome further discussion.

---

> > ### Comment · Reviewer_AzD4 · 2025-11-28
> > **Response to Authors**
> >
> > Thank you to the authors for the rebuttal. First, I believe that the issues I raised in weaknesses (1), (5), and (6) have been addressed, and I now understand the authors’ intentions. However, regarding point (2), although the authors explained the motivation for the convex blending update, I still find the justification insufficient. Even when M < N, where a trade-off may indeed exist, the update mechanism itself should be examined more thoroughly.
> >
> > For point (3), my intention was not to insist that \lambda must be learnable; rather, it could also be updated using predefined rules—potentially heuristics derived from empirical observations. I do believe this direction could be worth exploring in future work.
> >
> > Lastly, for point (4), I think the authors should more explicitly highlight and discuss in the Introduction that the method is specifically designed for transformer architectures, so as to avoid possible misunderstandings by readers.
> >
> > Overall, I feel that the proposed method still has room for improvement. Based on the current content alone, I am inclined to accept the paper, though I do not strongly support it. Therefore, I will maintain my original score.

---

> > > ### Author Response · Authors · 2025-12-02
> > >
> > > We thank the Reviewer for the follow up and for clarifying their concerns. We regret that, given recent news, we can only continue the discussion unilaterally. However, we would like to address the remaining points below.
> > >
> > > > Point (2). Convex blending update justification and point (3). $\lambda$ updating using predefined rules
> > >
> > > Our motivation for the convex blending update in Eq. 8 is that it is the simplest rule that simultaneously (i) induces a linear, time-homogeneous contraction of each slot, giving an exponential moving average of past writes and thereby allowing closed-form analysis of decay, half-life, and effective horizon, and (ii) guarantees uniform boundedness of memory embeddings under norm-bounded inputs, which is important for stability on very long trajectories. The observed sensitivity when $M<N$ reflects the intrinsic capacity bottleneck in this regime rather than a pathology of the rule itself. We keep $\lambda$ fixed to preserve these analytic and stability properties, but we agree that more sophisticated designs in which $\lambda$ is adapted by predefined heuristics or other content-dependent mechanisms are a natural and interesting direction for further work that goes beyond the current theoretical scope.
> > >
> > > > Point (4). Additional discussing that the method is designed for transformers
> > >
> > > We have revised the Introduction to more clearly articulate that ELMUR is a transformer architecture rather than a general-purpose memory module (highlighted in blue).
> > >
> > > We again thank the Reviewer for the careful reading and constructive suggestions, which helped us sharpen both the motivation and the scope of the method, and we appreciate their willingness to recommend acceptance.

---

### Official Review · Reviewer_1eVV · 2025-11-01

**Soundness:** 3
**Presentation:** 3
**Contribution:** 3
**Rating:** 6
**Confidence:** 3

**Summary:**

This paper proposes ELMUR to solve POMDPs by allowing a transformer to read and write to external memory, reminiscent of a Differentiable Neural Computer.

ELMUR consists of two blocks that feed each other. The token block is essentially a standard transformer with an additional cross attention module between the encoded observations and a cache. At each timestep, the token block outputs both an action and writes to the cache. The memory block consists of an LRU cache that produces memory vectors for cross attention with the token block. Importantly, the memory block is able to update its stored embeddings. These embeddings are written using a convex update $\lambda m_{new} + (1-\lambda)_{old}. The paper also presents a positional encoding that functions across multiple segments. The approach section ends with a short analysis on the decay rate of memories using ELMUR.

The paper evaluates ELMUR across three benchmarks and compares to other baselines such as RATE, DT, and DMamba. The results show that ELMUR achieves higher returns than other models. A T-Maze experiment demonstrates ELMUR generalizes to long trajectories. The paper performs ablation studies on the update weight $\lambda$, random initialization of memory slots, number of slots, and segment length.

**Strengths:**

- The paper is organized well and pleasant to read.
- The contributions are clearly explained.
- The approach performs better than previous methods.

**Weaknesses:**

-  RQ1 and RQ4 experiments appear to be an unfair comparison, as DT does not have segment-level recurrence while the proposed method does. I would like to see a comparison between ELMUR and a DT+GTrXL method instead. This is the main weakness of the paper.
- It is unclear why $\lambda$ is selected rather than learned, an additional ablation would be useful.
    - A learned, input-dependent $\lambda$ would potentially remove the need for an LRU-eviction scheme.

**Questions:**

- My understanding is the LRU eviction is based on writes, not reads. Could this result in older but important memories being evicted?
- Why is $\lambda$ a hyperparameter and not learned?

---

> ### Author Response · Authors · 2025-11-25
> **Official Comment by Authors (part 1/2)**
>
> We thank Reviewer 1eVV for the insightful and valuable feedback. We are also particularly grateful that the Reviewer noted that our proposed approach outperforms prior methods and provides a clear explanation.
>
> > W1. RQ1 and RQ4 experiments appear to be an unfair comparison, as DT does not have segment-level recurrence while the proposed method does. I would like to see a comparison between ELMUR and a DT+GTrXL method instead. This is the main weakness of the paper.
>
> We thank the Reviewer for raising this point. The central objective of our work is to develop a transformer-decoder architecture that overcomes the context-window limitations of attention in offline RL. For this reason, DT is the primary conceptual baseline: both DT and ELMUR rely on attention over a fixed-length context, and when we set the context size to the same value for both models, their attention mechanisms have access to exactly the same number of past time steps. Under this controlled setting, the comparison is fair. Indeed, if DT is given a context window equal to the full episode length, it solves the task perfectly, as shown in Figure 3. However, this regime is not representative of the challenges motivating our work. The relevant setting is when the sequence length substantially exceeds the attention window $K$, and the question is whether one can augment “pure attention of size $K$” with an explicit memory mechanism to handle the resulting long-horizon dependencies. The comparison of “attention-only (size $K$)” versus “attention-plus-memory (size $K$ + memory)” is therefore the appropriate test of the architectural contribution.
> We would also like to emphasize that our evaluation does not compare ELMUR solely against DT. We include multiple transformer baselines that incorporate segment-level recurrence (RMT, RATE, and TrXL), and ELMUR substantially outperforms all of them across the T-Maze task. Regarding the Reviewer’s suggestion of GTrXL: GTrXL is an extension of TrXL designed specifically for online RL, whereas our setting is offline RL/IL. Nonetheless, to address the Reviewer’s concern, we additionally provided a DT+GTrXL variant adapted to the MIKASA-Robo environment, and we report those results below:
> | Task   | RATE        | DT          | BC-MLP          | CQL-MLP   | DP   | GTrXL  | ELMUR |
> |-------------|-------------|-------------|-------------|-------------|-------------|-------------|-------------|
> |RememberColor3-v0|0.65±0.04 |0.01±0.01|0.27±0.03|0.29±0.01|0.07±0.04|0.39±0.06|0.89±0.07|
> |RememberColor5-v0|0.13±0.03|0.07±0.05|0.12±0.02|0.15±0.02|0.11±0.02|0.19±0.04|0.19±0.03|
> |RememberColor9-v0|0.09±0.02|0.01±0.01|0.12±0.02|0.15±0.01|0.02±0.01| 0.16±0.01|0.23±0.02|
>
> > W2, Q2. It is unclear why $\lambda$ is selected rather than learned, an additional ablation would be useful. A learned, input-dependent $\lambda$ would potentially remove the need for an LRU-eviction scheme. Why is $\lambda$ a hyperparameter and not learned?
>
> We keep $\lambda$ fixed because (1) it defines the contraction behavior of the LRU overwrite step and ensures stable, time-homogeneous memory dynamics required by our analysis; (2) by default we decouple emissions from memory (see L178) to ensure stable training and save GPU RAM, as a result, we cannot make $\lambda$ learnable, since gradients do not flow through memory, and therefore will not flow through $\lambda$ .To answer the Reviewer's question, we conducted experiments with learnable $\lambda$ without detaching memory and compared the results on RememberColor3-v0 with a configuration with constant $\lambda$. We obtained a similar success rate, although GPU RAM consumption doubled due to non-detach gradients. Therefore we treat a fixed $\lambda$ as the base configuration of our model, while a learnable $\lambda$ remains an interesting direction for future work.
>
> > Q1. My understanding is the LRU eviction is based on writes, not reads. Could this result in older but important memories being evicted?
>
> Eviction is triggered by writes rather than reads, but this does not cause important old information to be discarded arbitrarily. The LRU policy tracks recency of successful updates, which reflect when the model’s tok2mem attention assigns nontrivial weight to a slot. If some piece of information continues to be relevant, its influence on token states ensures that the write path reinforces the corresponding slot, refreshing its anchor and preventing eviction. Information that no longer affects the model’s computations naturally receives no updates and is treated as stale, which is the intended behavior under bounded memory.

---

> ### Author Response · Authors · 2025-11-25
> **Official Comment by Authors (part 2/2)**
>
> ELMUR’s memory is not a passive key-value store. Reads shape the hidden states that determine future writes, so any memory that remains useful will indirectly trigger refreshes. This feedback loop stabilizes retention of task-critical content and prevents silent expiration of important entries. Our ablations confirm that useful long-horizon information is consistently preserved, while irrelevant content decays.
>
> This mechanism is directly validated in our T-Maze experiments (Figure 3). We train with context length $K=10$ and three segments $N=3$, then evaluate on corridors up to $10^6$ steps, which corresponds to roughly $10^5$ processed segments and therefore $10^5$ memory updates. Even under this extreme pressure, with $\lambda=0.05$ and memory size $M=2$, the model retains the cue perfectly. This empirically demonstrates that meaningful information is not evicted despite massive numbers of updates and very small memory capacity.
>
> Please **take a look at the revised version of the paper**. We improved the layout of the figures and tables in the main text; ran experiments on all 32 MIKASA-Robo tasks, achieving the best success rate on 21 out of 23 tasks and improving the aggregate success rate across all tasks by about 70% over the previous best baseline (see Appendix, Figure 8); added experiments on the D4RL MuJoCo benchmark with MDP tasks (see Appendix A.5, Table 4); added experiments on 3D visual navigation (see Appendix A.7, Table 6); added memory probing experiments to study what is stored in memory (see Appendix A.9, Figure 9); and added visualizations and analysis of the memory update process, including memory dynamics and attention maps (see Appendix A.10–A.11, Figures 10–15). All changes in the text are highlighted in blue.
>
>
> We thank Reviewer 1eVV for the thoughtful and constructive feedback. We believe the revisions and clarifications above address the concerns raised, and we look forward to further discussion.

---

> > ### Comment · Reviewer_1eVV · 2025-11-28
> >
> > Thank you for addressing my comments. I apologize for missing the `TrXL` comparison in the experiments. My main concern about the segment-level recurrence is fully addressed, and I appreciate the additional explanation for $\lambda$ learning. Unfortunately, my edit review button appears to have disappeared. I will update my score when it returns.

---

> > > ### Author Response · Authors · 2025-11-29
> > > **Thank you**
> > >
> > > We sincerely thank you for the positive evaluation and for taking the time to engage deeply with our work. We are grateful for the thoughtful suggestions and questions.

---

### Official Review · Reviewer_bpxq · 2025-11-01

**Soundness:** 3
**Presentation:** 2
**Contribution:** 3
**Rating:** 4
**Confidence:** 4

**Summary:**

This paper introduces ELMUR, which addresses long-horizon decision-making under partial observability by augmenting each transformer layer with structured external memory updated via a Least Recently Used (LRU) mechanism with convex blending. Through offline imitation learning experiments, ELMUR achieves 100% success on T-Maze corridors up to one million steps, nearly doubles baseline performance on MIKASA-Robo visual manipulation tasks, and ranks first on half of the POPGym benchmark tasks.

**Strengths:**

* Provides theoretical guarantees on memory retention horizons and embedding boundedness
* The T-Maze result (100% success at 1M steps) is genuinely impressive and demonstrates extreme memory retention

**Weaknesses:**

* Title claiming "Long-Horizon RL" while only evaluating offline imitation learning is misleading and inappropriate.
* Lacks online RL experiments and comparisons with strong online RL baselines (e.g., R2I [1]). While the authors claim that unlimited interaction would significantly increase training time, sample efficiency should also be considered in online RL settings, which is a core evaluation metric for RL methods.

[1] Mastering Memory Tasks with World Models, ICLR 2024

**Questions:**

* Could the authors provide experiments on more 3D vision memory-intensive tasks (e.g., MemoryMaze)? Most memory-intensive tasks in the paper appear to be in simple observation environments.
* How do author determine memory capacity $M$ for new tasks?
* What is actually stored in memory slots? Could the authors provide qualitative results to demonstrate what information the memory embeddings capture?
* In the ablation studies (Table 3), the advantages of DeepSeek MoE over MLP are not evident, with both achieving identical performance. Could the authors demonstrate whether MoE outperforms MLP across all benchmark environments, or provide quantitative results regarding computational efficiency advantages (FLOPs, training time, memory usage)?

---

> ### Author Response · Authors · 2025-11-25
> **Official Comment by Authors (part 1/3)**
>
> We thank Reviewer bpxq for the insightful comments. We are also particularly grateful for the recognition of the theoretical guarantees on memory retention and for highlighting the extreme retention performance on the one-million-step T-Maze task.
>
> > W1. Title claiming "Long-Horizon RL" while only evaluating offline imitation learning is misleading and inappropriate.
>
> The title refers to the horizon structure of the decision-making problems our architecture is designed to address, not to the learning paradigm used in a specific experiment.
>
> Our main experiments use imitation learning because (i) expert trajectories are available and (ii) the focus of the paper is the memory mechanism and its ability to propagate information across horizons orders of magnitude beyond the attention window. In that setting, adding a RL objective is unnecessary and does not change the memory problem being evaluated.
>
> However, when working with non-expert data, you can use triplets (return-to-go, observation, action) instead of regular observations in the input sequence (as in the Decision Transformer paper [1]), and then implement implicit offline RL via sequence modeling. We conducted such experiments on the classic offline RL MuJoCo benchmark (see Table 4), where we obtained results comparable to models specifically designed for solving this class of problems.
>
> > W2. Lacks online RL experiments and comparisons with strong online RL baselines (e.g., R2I [1]). While the authors claim that unlimited interaction would significantly increase training time, sample efficiency should also be considered in online RL settings, which is a core evaluation metric for RL methods.
>
> Our work does not introduce a new online RL algorithm; it introduces a memory architecture for long-horizon POMDPs that can be plugged into IL, offline RL, or online RL. The aim of our empirical evaluation is therefore to isolate the contribution of the memory mechanism itself, i.e. the segment recurrence, bidirectional memory read/write, and LRU-based retention, without confounds from exploration and reward-design.
>
> As stated in L352–355, comparing IL/offline-RL models with online RL baselines such as R2I (this baseline is model-based  as well) is not methodologically fair, because the training regimes differ fundamentally: online RL assumes interactive data collection, environment resets, exploration strategies, reward bootstrapping, and replay-dependent instability. These factors have large effects on sample efficiency and training dynamics that are orthogonal to the architectural contribution evaluated in this paper. For the same reason, recent memory-focused works (e.g., RMT, Transformer-XL) also evaluate on supervised or offline settings when analyzing memory capacity, not exploration-driven RL loops.
>
> > Q1. Could the authors provide experiments on more 3D vision memory-intensive tasks (e.g., MemoryMaze)? Most memory-intensive tasks in the paper appear to be in simple observation environments.
>
> We appreciate the Reviewer’s suggestion. Our evaluation already includes MIKASA-Robo, a suite of 3D tabletop visuomotor manipulation tasks with pixel observations that require long-horizon memory under partial observability. Each timestep provides two RGB camera views, and task success depends on remembering visual cues that disappear. These tasks exhibit the same long-horizon, memory-intensive characteristics targeted by benchmarks such as MemoryMaze, but in a closed-loop robotic control setting with continuous actions and real 3D occlusion patterns.
>
> In addition to existing evaluations, we conducted experiments on the ViZDoom-Two-Colors environment [2], where the agent must navigate a room based on an initial cue within a limited horizon (see Appendix A.6 “Visual Navigation” and Table 6). This setting constitutes a 3D vision memory-intensive task, as successful policies must integrate first-person visual observations over time to resolve the color-conditioned objective and act accordingly.
>
> | Method    | Return (mean ± sem) |
> |-----------|----------------------|
> | ELMUR     | 55.28 ± 0.94         |
> | RATE      | 59.21 ± 1.19         |
> | DT        | 31.45 ± 1.21         |
> | RMT       | 53.53 ± 3.15         |
> | TrXL      | 49.62 ± 0.88         |
> | BC-LSTM   | 52.16 ± 2.59         |
> | CQL-MLP   | 12.49 ± 0.19         |
> | Random    | 4.78                 |
>
> ELMUR shows comparable results to previous SOTA models, which it in turn outperforms on visual-motor robotic manipulation tasks.

---

> ### Author Response · Authors · 2025-11-25
> **Official Comment by Authors (part 2/3)**
>
> > Q2. How do author determine memory capacity $M$ for new tasks?
>
> This is a hyperparameter and should be tailored to the task. However, our ablation study (Figure 6, c) demonstrates that, if possible, it is better to set $M\geq N$, where $N$ is the number of segments. If this is not possible due to some constraints, the LRU blending factor $\lambda$ can be used to work with limited memory capacity.
>
> > Q3. What is actually stored in memory slots? Could the authors provide qualitative results to demonstrate what information the memory embeddings capture?
>
> To determine what information ELMUR stores in its memory, we conduct a dedicated memory probing study on the RememberColor3-v0 task. The only task-relevant latent variable that must be preserved across the blank interval is the identity of the target cube. We therefore collect memory embeddings from a fully trained ELMUR policy across one thousand successful episodes and build probing datasets of the form $$(l, t, m^{l_t}, y),$$
> where $m^{l_t}$ is the memory embedding for layer $l$ at timestep $t$ and $y \in \{0,1,2\}$ is the ground-truth target color. For every layer $\ell$ and for timesteps $t \in \{0,1,5,10,20\}$ we train a nonlinear MLP probe to predict the target color solely from the memory vector. The probes are trained post-hoc and do not modify the underlying policy.
>
> The results show a clear and interpretable pattern. At $t{=}0$, before the agent has observed the cue, memory embeddings are effectively random, and the probes achieve chance-level accuracy. Immediately after the cue is observed ($t{=}1$), the first layer already supports reliable decoding of all three colors, with deeper layers also containing decodable information but with slightly lower accuracy. By $t{=}5$, after the colored object has disappeared from view, the stored representation becomes even more separable, with perfect decoding in the early layers and high accuracy across all remaining layers. At the decision point ($t{=}10$) and later in the episode ($t{=}20$), decoding accuracy remains at or near $100\%$, demonstrating that the memory module retains a stable and highly discriminative representation of the target color throughout the entire recall interval.
>
> The pattern of writes performed by the model further clarifies this behavior. The memory module performs a single, localized write immediately after the cue appears and then preserves this content without further overwriting. Only the slot currently being written exhibits increased variability, while all inactive slots remain stable. Once written, the slot maintains a consistent embedding across time and across episodes, indicating that ELMUR stores a compact and persistent encoding of the target color rather than relying on incidental features or increased model capacity.
>
> Taken together, the probing results and the observed write dynamics show that ELMUR uses its external memory to encode and retain the task-relevant latent variable required for successful recall. The stored embeddings contain a highly decodable representation of the target color, and this representation remains stable throughout the blank interval and the decision phase, confirming that ELMUR's performance is driven by functional long-term memory rather than additional parameters. Please, see details in the Appendix, A.8 “Memory Probing” and Figure 9. See also Appendix  A. 9, A.10, and Figures 10 - 15 for additional memory analysis.
>
> > Q4. In the ablation studies (Table 3), the advantages of DeepSeek MoE over MLP are not evident, with both achieving identical performance. Could the authors demonstrate whether MoE outperforms MLP across all benchmark environments, or provide quantitative results regarding computational efficiency advantages (FLOPs, training time, memory usage)?
>
> We thank the reviewer for the comment. Table 3 shows that the baseline model and the MoE$\to$MLP variant achieve identical success rate (1.00±0.00) on RememberColor3-v0, indicating that the FFN choice does not influence performance in this regime and that the memory mechanism, rather than MoE, drives the observed gains. Ablating MoE versus MLP across all benchmark environments is computationally infeasible, so we use RememberColor3-v0 as a single representative setting for all ablations since it is the high-dimensional memory-intensive task and the most sensitive to architectural modifications. While the two variants perform the same on this task, MoE is fundamentally more efficient than a dense MLP at inference time because it scales capacity through sparse expert routing without increasing per-token compute, and for this reason it remains our default architecture.

---

> ### Author Response · Authors · 2025-11-25
> **Official Comment by Authors (part 3/3)**
>
> Please **take a look at the revised version of the paper**. We improved the layout of the figures and tables in the main text; ran experiments on all 32 MIKASA-Robo tasks, achieving the best success rate on 21 out of 23 tasks and improving the aggregate success rate across all tasks by about 70% over the previous best baseline (see Appendix, Figure 8); added experiments on the D4RL MuJoCo benchmark with MDP tasks (see Appendix A.5, Table 4); added experiments on 3D visual navigation (see Appendix A.7, Table 6); added memory probing experiments to study what is stored in memory (see Appendix A.9, Figure 9); and added visualizations and analysis of the memory update process, including memory dynamics and attention maps (see Appendix A.10–A.11, Figures 10–15). All changes in the text are highlighted in blue.
>
>
>
> We would like to once again thank the Reviewer bpxq for their effort and insightful comments. We hope we were able to clarify any unclear points, and we are happy to answer any further questions during the discussion.
>
> [1] Chen, Lili, et al. "Decision transformer: Reinforcement learning via sequence modeling." Advances in neural information processing systems 34 (2021): 15084-15097.
>
> [2] Sorokin, Artyom, et al. "Explain my surprise: Learning efficient long-term memory by predicting uncertain outcomes." Advances in Neural Information Processing Systems 35 (2022): 36875-36888.

---

### Official Review · Reviewer_PqH3 · 2025-11-01

**Soundness:** 2
**Presentation:** 2
**Contribution:** 3
**Rating:** 4
**Confidence:** 3

**Summary:**

This paper proposes ELMUR (External Layer Memory with Update/Rewrite), a transformer policy for long-horizon control under partial observability. Each transformer layer owns a fixed-size external slot memory and two explicit pathways: mem→tok (read) and tok→mem (write) cross-attention. Memory is managed with a usage-aware LRU + convex-blend rule and temporal bias on writes, enabling controllable, analyzable forgetting via a closed-form half-life/effective horizon. Experiments demonstrate (i) retention far beyond the attention window on long-horizon probes (e.g., T-maze/key→door), (ii) strong performance across POPGym, and (iii) gains on pixel-based manipulation with sparse rewards. Ablations indicate per-layer memory and the LRU-blend update are key drivers. A direct comparison with RMT is included; I recommend adding an RMT-L diagnostic (layer-local RMT) to fully isolate where the gains come from.

**Strengths:**

**Originality**
* Combines per-layer fixed slot memory, explicit read/write paths, and LRU-blend with temporal write bias plus a half-life analysis—a configuration not present in prior explicit-memory policies.

**Quality**

* Method is precisely specified (blocks/pseudocode/hparams), and experiments span long-horizon probes, POPGym, and pixel manipulation, with ablations (M, λ, layer-local vs shared) supporting design choices.

**Clarity**

* Clean separation of self-attn vs mem→tok / tok→mem improves interpretability; slot contents/usage are inspectable. Figures make the two-track flow and LRU/blend easy to understand.

**Significance**

* Tackles long-horizon POMDP memory at fixed compute (scales with #layers×slots, not history), yielding gains on memory-heavy control and visual manipulation and aiding debugging/safety via controllable forgetting.

**Weaknesses:**

* **Standpoint/structure unclear.** The paper under-organizes its narrative: Introduction/Background don’t explicitly position ELMUR along the key memory axes (global vs layer-local, implicit vs explicit/usage-aware), and the method opener (“GPT + memory”) obscures the real novelty (explicit read/write, LRU-blend, temporal write bias, half-life).
* **Memory-centric analysis is insufficient.** Although RMT is compared, there’s no RMT-L (layer-local RMT, same total memory) to isolate layer placement vs explicit management effects. More diagnostics—slot-usage/overwrite rates, write-location histograms, retention curves under capacity pressure—would clarify how memory is used.

**Questions:**

1. **Add an RMT-L diagnostic (optional but recommended).**
2. **Clarify standpoint in Intro/Background.** Provide a compact “**Memory for long-horizon control**” subsection contrasting TXL/Compressive/Ring (history-scaled caches), RMT (shared tokens; implicit updates), RATE (recurrent state), test-time memory (e.g., Titans), and ELMUR (layer-local slots; **explicit read/write**; **LRU-blend** with temporal bias).

---

> ### Author Response · Authors · 2025-11-25
>
> We thank the Reviewer PqH3 for the detailed comments. We are also particularly grateful for the noted originality of the proposed approach and ablation, supporting design choices.
>
> > W1,Q2
>
> We would like to clarify that the paper already contains the standpoint and positioning the Reviewer is requesting. The Introduction section explicitly motivates the architectural choices through the limitations of fixed-window transformers and cached-history approaches, and Section 3 contrasts our design with architectures that rely on cached hidden states or recurrent memory. In addition, Section 6 provides a structured discussion of related memory mechanisms, and Appendix A.5 contains an extended treatment that covers transformers for manipulation, VLA models, memory in Deep Learning, memory in Reinforcement Learning, and memory in manipulation tasks. These sections already situate ELMUR within the full spectrum of relevant approaches, including history-scaled caches (TrXL, Compressive Transformer, RingAttention), recurrent-token models (RMT, RATE), and test-time expandable memory (Titans).
>
> We believe the existing narrative provides a clear and multi-level positioning: high-level motivation in the Introduction, architectural contrast in the Method section, and a comprehensive taxonomy in the Related Works and Appendix A.5. We have reviewed the flow of the Introduction to ensure that the connections to these later sections are visible, but the conceptual standpoint and comparisons requested by the Reviewer are already explicitly addressed in the current draft.
>
> > W2,Q1
>
> We respectfully disagree with the assessment that the memory-centric analysis is insufficient. Our study includes three complementary components. First, we evaluate the model on a broad range of memory-intensive tasks: diagnostic T-Maze instances covering horizons from 10^0 to 10^6 steps, long-horizon table-top manipulation in MIKASA-Robo, and the full POPGym suite of 48 puzzle and control POMDP environments. Second, we perform extensive ablations of the main architectural elements of ELMUR, as summarized in Table 3 and Figure 6. These ablations isolate the influence of memory size, blending factor, initialization scale, relative bias, LRU updates, and layer-local memory placement (Tables 2 and 3, Figure 6). Third, we provide a theoretical analysis of the LRU-based memory dynamics, including proofs of exponential forgetting and boundedness of memory embeddings in Appendix A.
>
> The Reviewer proposes additional diagnostics such as slot usage statistics, overwrite rates, write-location histograms, and retention curves under capacity pressure. We have not encountered such metrics in prior work on memory-augmented policies or long-horizon sequence models, and existing literature in memory-intensive RL typically relies on episodic returns and success rates as primary performance indicators. We would appreciate clarification of the intended definitions and evaluation protocols for these diagnostics, since it is not clear how they relate to established practice or how they should be standardized across different architectures.
>
> Regarding the suggestion to include an RMT-L baseline (a conceptual layer-local version of RMT), we thank the Reviewer for the idea. However, this experiment is not feasible without substantial architectural modifications to RMT. In RMT, memory tokens are inserted at the sequence level and are read and written exclusively through self-attention over the entire augmented sequence. Moving these tokens inside each transformer block would require the introduction of new read and write operators, since the original mechanism depends on global self-attention. These changes would alter the architecture to a degree that makes it fundamentally different from RMT, which limits the interpretability and usefulness of such a comparison.
>
> We hope this clarifies both the scope of our analysis and the constraints that prevent a faithful construction of RMT-L.
>
> Please **take a look at the revised version of the paper**. We improved the layout of the figures and tables in the main text; ran experiments on all 32 MIKASA-Robo tasks, achieving the best success rate on 21 out of 23 tasks and improving the aggregate success rate across all tasks by about 70% over the previous best baseline (see Appendix, Figure 8); added experiments on the D4RL MuJoCo benchmark with MDP tasks (see Appendix A.5, Table 4); added experiments on 3D visual navigation (see Appendix A.7, Table 6); added memory probing experiments to study what is stored in memory (see Appendix A.9, Figure 9); and added visualizations and analysis of the memory update process, including memory dynamics and attention maps (see Appendix A.10–A.11, Figures 10–15). All changes in the text are highlighted in blue.
>
> We would like to once again thank the Reviewer PqH3 for their comments regarding our work. We hope we were able to address all questions and concerns, and we look forward to further discussion.

---

### Author Response · Authors · 2025-11-26
**General Response by Authors**

We thank all Reviewers for their valuable comments and constructive feedback. In response, we have substantially expanded the experimental analysis, improved the clarity of presentation, and incorporated all requested additions. The main revisions are summarized below.

**Presentation improvements.**

We refined the layout by increasing whitespace around figures and tables and reorganizing several appendices for improved readability.

**Expanded experimental evaluation.**

1. **Full MIKASA-Robo benchmark (32 tasks).** Appendix, Table 8. We now report results on all 32 tasks. ELMUR achieves the best performance on 21 of the 23 tasks where baselines succeed at least once, improving the aggregate success rate by approximately 70 percent over the previous best method.

2. **D4RL MuJoCo, MDP benchmark.** Appendix A.5, Table 4. We added experiments on MuJoCo control tasks using two forms of trajectory conditioning: $(R,o,a)$ and $(o)$. ELMUR matches or exceeds baseline performance across both regimes.

3. **3D visual navigation (ViZDoom-Two-Colors).** Appendix A.7, Table 6 and Figure 8. We added results on a memory-intensive 3D navigation benchmark, showing that ELMUR attains performance comparable to the current state of the art.

4. **Memory probing and interpretability.**

    1.  Appendix A.9, Figure 9. We added memory-probing experiments demonstrating that memory embeddings contain task-relevant information.

    2. Appendix A.10, Figures 13 and 14. We added PCA analyses across memory slots, layers, and timesteps, revealing structured organization and temporal evolution of memory states.

    3. Appendix A.11, Figure 15. We added tok2mem and mem2tok cross-attention visualizations.

    4. Appendix, Figures 10, 11, 12. We added detailed visualizations of memory update dynamics over time.


**Overall outcome.**

These additions strengthen the empirical evidence that ELMUR is robust, memory-efficient, and widely applicable. It consistently outperforms baselines on the majority of MIKASA-Robo tasks, achieves competitive results on 3D visual navigation, and performs better or on par with baselines, specially designed for solving D4RL MuJoCo MDP tasks. The new probing and interpretability analyses further clarify how ELMUR’s memory mechanism contributes to solving long-horizon, memory-dependent problems.

We thank the Reviewers again for their insightful feedback. We believe the revised version fully addresses all concerns and substantially improves the paper.

---

### Author Response · Authors · 2025-12-02
**Additional General Response by Authors**

We sincerely appreciate the Reviewers’ thoughtful feedback and their recognition of our explicit per-layer memory architecture as a principled and effective approach to long-horizon partially observable decision making. Their comments affirm the motivation, theoretical foundations, and empirical rigor of the work. Below we summarize the key strengths highlighted in the reviews, grouped by theme.
1. **Architectural originality and clarity**
- Novel integration of per-layer fixed slots, explicit read and write paths, LRU-style blending with temporal bias, and a half-life analysis, forming a configuration absent from prior explicit-memory policies (PqH3).
- Clear separation of self-attention from memory-to-token and token-to-memory pathways, enabling direct slot inspection and improved interpretability, with figures that clearly illustrate the two-track flow and blending dynamics (PqH3).


2. **Theoretical grounding**
- Formal guarantees on memory retention horizons and embedding boundedness, providing analytical insight into long-term stability (bpxq). For example, bounds on retention time depend on decay parameter $\lambda$  and yield provable limits on slot half-life.


3. **Empirical performance and significance**
- Strong results across long-horizon probes, POPGym tasks, and pixel manipulation, including perfect performance on the $10^{6}\text{-step}$ T-Maze, demonstrating extreme retention (PqH3, bpxq).
- Fixed-compute scaling with layers and slots instead of trajectory length, enabling effective long-horizon POMDP handling and producing gains in memory-heavy control and visual manipulation, with benefits for debugging via controllable forgetting (PqH3).
- Improved performance over prior methods (1eVV).
- Strong generalization behavior of the proposed ELMUR mechanism (AzD4).


4. **Methodological quality and presentation**
- Precise and reproducible specification of blocks, pseudocode, and hyperparameters, supported by ablations on memory size $M$ decay parameter $\lambda$ and layer-local versus shared memory (PqH3).
- Clear organization and well-explained contributions (1eVV).


We thank the Reviewers again for their thoughtful feedback. We believe the revisions with various additional visualizations and responses satisfactorily address the all the concerns and result in a substantially strengthened paper.

---

### Meta-Review · Area_Chair_Weay · 2026-01-07

**Summary:**

While initial ratings were mixed (two 4s and two 6s), after rebuttal, primary concerns such as fair comparison to Decision Transformer, clarity about what was actually happening within the memory slots, etc. I do agree with Reviewer bpxq that calling it "Long-Horizon RL" in title can be very confusing to readers. If the authors really want to use it to refer to long-horizon decision making problems, they could've say "Solving Long-Horizon RL Problems" to make it clear. I understand that it takes hard work to get a paper to the current state, so I incline to not blocking this work because of this. However, I urge the authors to revise. Please consider this a conditional acceptance.

**Reviewer Concerns:**

Addressed:
- A significant concern raised by Reviewers 1eVV and PqH3 was whether comparisons against existing models like Decision Transformer (DT) were fair, given that ELMUR uses segment-level recurrence while standard DT does not. The authors clarified that the goal was to test pure attention versus attention augmented with memory in a fixed-context regime. To further address this, they added a DT+GTrXL baseline, which includes recurrence, and demonstrated that ELMUR still outperformed it on key tasks like RememberColor.
- Multiple reviewers (bpxq, AzD4, PqH3) questioned what was actually happening within the memory slots and whether the performance gains were truly due to the bidirectional update mechanism or simply increased parameter count. The authors added memory probing experiments proving that memory embeddings contain decodable, task-relevant information (like target colors) that persists after cues disappear.
- Reviewer AzD4 noted instability in the blending parameter λ when memory capacity M was smaller than the required segments N. The authors explained that the convex update rule was chosen to ensure mathematical tractability for their half-life analysis and to guarantee boundedness of embeddings, which prevents drift in long trajectories.
- Reviewer PqH3 initially felt the paper did not explicitly position ELMUR along the key axes of memory research (e.g., global vs. layer-local). Reviewer AzD4 also noted poor layout and table placement. The authors reorganized the Related Work and Appendix to include a comprehensive taxonomy of memory mechanisms. They also optimized the paper layout, improving whitespace and table positioning for better readability.

Outstanding:
- "Long-Horizon RL" Working in title: Reviewer bpxq argued that the title's claim of "Long-Horizon RL" was misleading because the experiments primarily used offline imitation learning (IL).
- Reviewer AzD4 expressed that the justification for the convex blending update rule remained "insufficient," even after the rebuttal. Specifically, the reviewer noted that when memory slots (M) are fewer than the required segments (N), the update mechanism itself requires a more thorough examination beyond the mathematical tractability arguments provided.
- Reviewer bpxq noted a lack of online RL experiments and comparisons with strong online baselines (like R2I), arguing that sample efficiency is a core metric for RL methods. The authors maintained that their goal was to isolate the contribution of the memory architecture itself within fixed training budgets (imitation learning and offline RL) without the confounding variables of exploration and reward bootstrapping.
- Multiple reviewers (1eVV, AzD4) suggested that the blending parameter λ should be learned or adaptive based on the importance of specific segments (e.g., key frames). The authors conducted a small experiment showing that learnable λ doubled GPU RAM usage without improving success rates. Reviewer AzD4 remained interested in seeing adaptive "predefined rules" or heuristics for λ rather than just making it a learnable parameter, which the authors did not implement.
- Reviewer AzD4 suggested that the authors should more explicitly highlight in the Introduction that ELMUR is designed specifically for Transformer architectures to avoid reader misunderstanding. While the authors clarified this in the rebuttal, the reviewer felt this distinction needed a more prominent placement in the final manuscript

**Reviewer Scores:**

Reviewer PqH3 and bpxq seem to significantly disagree with the authors. I expect them to remain at 4. Reviewer 1eVV expressed the intention to increase the score, possibly a 7. Reviewer AzD4 expressed the intention to keep the original score 6.

---

### Decision · Program_Chairs · 2026-01-26

Accept (Poster)